# Schema-based predictive eye movements support sequential memory encoding

Jiawen Huang[1]*, Isabel Velarde[1], Wei Ji Ma[2], Christopher Baldassano[1]

[1]Department of Psychology, Columbia University, New York, United States; [2]Center for Neural Science and Department of Psychology, New York University, New York, United States

**Abstract** When forming a memory of an experience that is unfolding over time, we can use our schematic knowledge about the world (constructed based on many prior episodes) to predict what will transpire. We developed a novel paradigm to study how the development of a complex schema influences predictive processes during perception and impacts sequential memory. Participants learned to play a novel board game ('four-in-a-row') across six training sessions and repeatedly performed a memory test in which they watched and recalled sequences of moves from the game. We found that participants gradually became better at remembering sequences from the game as their schema developed, driven by improved accuracy for schema-consistent moves. Eye tracking revealed that increased predictive eye movements during encoding, which were most prevalent in expert players, were associated with better memory. Our results identify prediction as a mechanism by which schematic knowledge can improve episodic memory.

## Editor's evaluation

This work designed a novel board game paradigm (4-in-a-row, a sort of tic-tack-toe expansion) in combination with eye movement recordings to examine how schemas gradually learned in the game influence memory. They provide impressive evidence that, in both behavior and eye movements, schema indeed guides the encoding of sequential events and facilitates memory performance. This paper would be of great interest to many fields, such as memory, learning, and decision-making.

**\*For correspondence:**
jh4290@columbia.edu

## Introduction

A key benefit of having a memory system is that it enables us to use the past to make predictions about the future, which is adaptive for survival (*Cowan et al., 2021*). Prediction can be defined as a top-down process in which people generate expectations about what they will experience next, based on their previous experiences and the current context. A particularly important source of prediction is schemas, which are adaptable knowledge structures reflecting generalized information abstracted from multiple episodic experiences (*Ghosh and Gilboa, 2014*). For example, we may have schemas about the kinds of objects that tend to occur at a beach, the social norms for ordering food at different kinds of restaurants, or the typical stages of a chess game.

Previous research has shown that having a detailed and robust schema yields improvements in memory (*Alba and Hasher, 1983*). This memory improvement has been attributed in part to processes during recall since schemas provide cues that can be used to retrieve episodic details that would otherwise be forgotten (*Anderson and Pichert, 1978*; *Watkins and Gardiner, 1979*). Additionally, schemas could play a role during memory encoding, by helping people represent information more meaningfully (*Bransford and Johnson, 1972*; *Chase and Simon, 1973*).

One question that has been relatively unexplored in this literature is whether schemas improve episodic memory by enabling sophisticated predictions during encoding. In the past 20 years, there has been a growing recognition of the importance of prediction on how we perceive, understand, and interact with the world (*Bar, 2009*; *Clark, 2013*; *Friston, 2010*; *Rao and Ballard, 1999*). There is evidence that domain experts (i.e. people with a strong schema in a specific field) automatically engage these predictive processes; for example, basketball experts shown a clip from a game tended to remember the final positions of the players as being ahead of their actual positions (*Gorman et al., 2012*). These predictive processes are only possible with a pre-existing schema and might partially explain the large literature showing selective memory improvement for schema-consistent (i.e. predictable) information (*Anderson, 1981*) and highly unexpected information (i.e. prediction errors, *Bonasia et al., 2018*; *Quent et al., 2021*).

Previous prediction research has typically used two types of approaches. The first approach uses people's pre-existing real-world knowledge, such as the regularity of language structure (*Goldstein et al., 2022*; *Shain et al., 2020*) or event sequences in a familiar setting (*Baldassano et al., 2018*). Due to their complexity, however, it is difficult to build a ground-truth model for these schemas. These types of knowledge are also learned slowly, making it difficult to study their development in a lab setting. The second paradigm, commonly used in memory research, teaches participants simple and novel sequences of discrete stimuli like abstract shapes and pictures (e.g. *Schapiro et al., 2012*; *Sherman and Turk-Browne, 2020*). Due to their simplicity, it is possible for participants to learn these sequences in a short period of time and they are easy to model. However, the predictive processes examined in these studies might not engage the same mechanisms as when predictions are based on more complex schemas that generalize to new stimuli.

In this study, we investigate prediction as a potential mechanism for schema-related memory improvement, in a domain that avoids the issues with overly simplistic or poorly defined schemas. Specifically, we have participants remember sequences from a simple board game recently developed by *van Opheusden et al., 2021* called four-in-a-row. In this two-player game, a generalization of tic-tac-toe, players compete to be the first to connect four pieces in a row on a 4 × 9 board. The schema of the game encompasses not just this rule (which is easily learned), but an understanding of what kinds of move sequences are typical, an emergent property of the game rules that requires playing experience to learn. Expertise in four-in-a-row has been shown to change how individual game boards are remembered, with experts better representing arrangements of pieces that are strategically important (*van Opheusden et al., 2021*). The game has a complexity far exceeding typical tasks in previous schema and prediction research, yet it is possible to capture near-optimal play with a linear model. The novelty and simplicity of the game ensure that participants start the experiment without a schema but can acquire the schema over several hours of practice, allowing us to tractably study how schema development is related to changes in prediction and memory longitudinally in a lab setting. The fact that possible moves correspond to different spatial locations allows us to use eye movements as an indication of people's predictive processes, as in previous paradigms with spatial actions (e.g. *Tal et al., 2021*).

Based on findings in previous schema research, we hypothesized that people's memory for move sequences would improve over training sessions, alongside improvements in a schema, operationalized as improved gameplay ability. Although previous research has sometimes found novelty-driven memory improvements for schema-inconsistent information (reviewed in *Frank and Kafkas, 2021*), studies of memory for complex memoranda such as chess boards have shown an advantage for schema-consistent stimuli (e.g. board positions from actual chess games) (*Gobet and Waters, 2003*). Thus, we hypothesized that the memory improvement resulting from the development of schema in four-in-a-row should similarly be specific to moves that are schema-consistent. People's schema quality should also be related to their memory performance, such that people with stronger gameplay abilities will have better sequence memory. If the prediction is indeed a potential mechanism for schema-related improvement, we would expect people's eye movements to become more anticipatory as they gain more experience playing the game and can rely more on internal predictive models. The extent of predictive eye movements should also mediate the relationship between schema quality and memory performance, providing a mechanism through which schematic knowledge can impact episodic memory. Here, we find support for all these hypotheses.

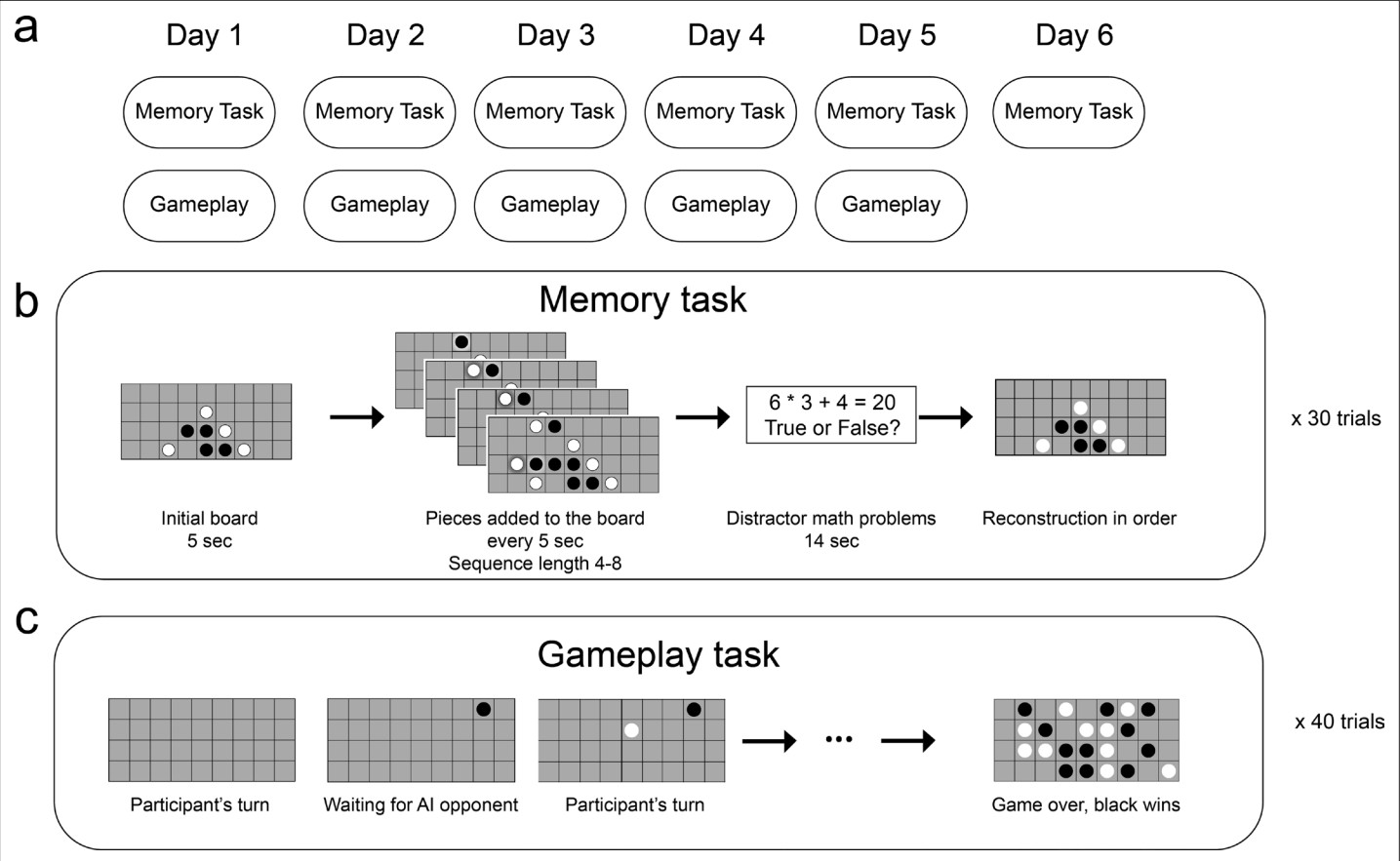

**Figure 1.** A schematic of the experimental method. (**a**) Task structure across six (non-consecutive) days. (**b**) Memory task. In each of the 30 trials, participants saw an initial board for 5 s, and then a move was added to the board every 5 s. After viewing a sequence of 4four to eight moves and completing a distractor task, participants were shown the initial board and asked to reconstruct the sequence by placing the pieces on the board. (**c**) Gameplay task. Participants played 40 games against an adaptive-difficulty AI agent.

## Results

### Memory and gameplay improvement

Over six sessions (each separated by 2.15 days on average), participants performed two separate tasks (*Figure 1*). In the memory task, participants were presented with a novel gameplay sequence and asked to remember it. After a distractor task, they were shown the first board of the sequence and they were given an unlimited amount of time to reconstruct the rest of the sequence from memory. In the first session, participants were not told that these stimuli came from a game, and were only instructed to remember circles appearing on a grid. The first session therefore provided a no-schema baseline, since participants could not use a game model to make predictions about upcoming moves. In a post-task questionnaire, we confirmed that in session 1 most participants did not suspect that the stimuli were from a game or guess the rules of the game (14 out of 19 in the online study, and 13 out of 16 in the in-person study). In a separate gameplay task (occurring after the memory task on all but the last day), they were provided with the rules of the game and played the four-in-a-row game against an AI opponent, staircase to match the skill level of the player.

Participants recall accuracy (combined across the online and in-lab data), calculated as the percent of moves they recalled at the right location in the right order, improved across training sessions, improving from 61.6% (SD = 18.1%) in session 1 to 69.5% (SD = 20.6%) in session 6 (*Figure 2a*). At the same time, participants' playing strength, measured with an Elo rating (*Elo, 1978*), increased (*Figure 2b*). Elo ratings are computed based on how often people win against opponents of varying skill levels, and we use Elo as a measure of schema quality – the better a player is, the better knowledge they have about the move probabilities during near-optimal gameplay. We fit mixed-effect models

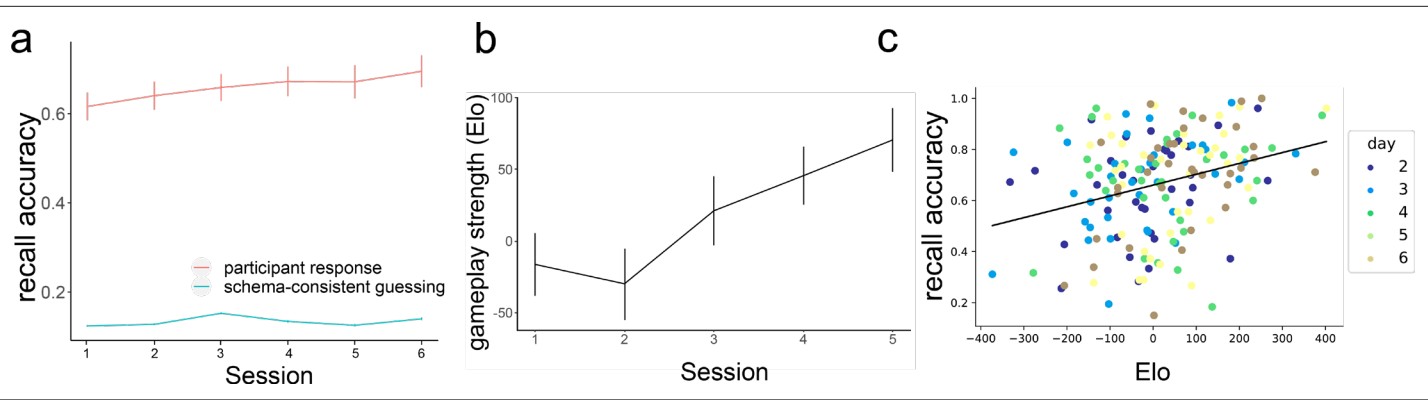

**Figure 2.** Participants' performance in memory and gameplay. (**a**) On average, Participants become better at remembering sequences over the course of the training (red line, N = 35). In each session, memory accuracy is much higher than the performance that would be achieved if people were simply guessing according to the gameplay model (green line, simulated, N = 100). (**b**) On average, participants become better at playing the game across sessions. Error bars represent the standard error of the mean. (**c**) There is a positive correlation between people's playing strength and their recall accuracy (each dot corresponds to one session of one participant).

with a fixed effect of the session and a random slope for the participant to predict recall accuracy and Elo. The fixed effect of the session was significant in both models (for memory, $\beta$=0.015, $t$=3.215, p=0.003; for Elo, $\beta$=24.92, $t$=5.102, p<0.001), demonstrating improvement in both recall accuracy and gameplay over time. We also found that Elo's rating in a session was correlated with memory performance in the following session, Pearson $r$=0.298, p<0.001 (*Figure 2c*), showing that having a better schema for the game was associated with better episodic memory for move sequences. To understand whether this relationship was present within individual participants, we fit a linear mixed-effect model to predict memory performance from Elo with per-participant intercepts and slopes as random effects. We found that the relationship between Elo and memory accuracy was not significant in this model ($\beta$=0.012, $t$=1.196, p=0.242), suggesting that this effect was primarily driven by individual differences (people with better schema tend to have better memory) rather than across-session improvements in Elo.

Out of the 30 sequences shown to participants in each session, 10 of them ended with one player successfully getting four pieces in a row (more details in Methods). After learning the rules of the game, the presence of such a win could be a salient event for the participant and could lead to changes in the memory performance. Indeed, we found that in sessions 2–6, memory for sequences that ended in a win state was significantly better ($t$=6.67, p<0.001). We did not observe this pattern in the first session (before participants were taught the game rules) and actually found a marginally significant effect in the opposite direction, with worse memory for winning sequences ($t$=–2.01, p=0.052). This result provides additional evidence that schema-related features of a sequence play a role in the memory performance.

## Modeling schema-related memory improvement

To investigate how schema consistency is related to memory for individual moves and how this relationship evolves, we trained a model of the schema for the game on moves played by a near-optimal AI agent. The model identifies the features each move forms (such as three-in-a-row, a line of three pieces of the same color) and assigns a value to each feature based on the training data (described in more detail in the Method section). Given a current board position, the model outputs a probability distribution over the next move that would likely be played by a very strong player (*Figure 3a*). Using this model, we can measure the extent to which each move shown to participants is likely (schema-consistent) vs unlikely (schema-inconsistent), and how this is linked to recall accuracy. Here, schema consistency is used as an objective, the subject-independent measure of how good a move is. Each subject will exhibit different degrees of alignment to this 'ground-truth' schema.

Separately for each session, we ran a mixed-effects logistic regression with subsequent memory (right or wrong) as the outcome variable, with the probability of the move as the fixed effect, and with a participant random intercept. As shown in *Figure 3b*, there was initially no relationship between the

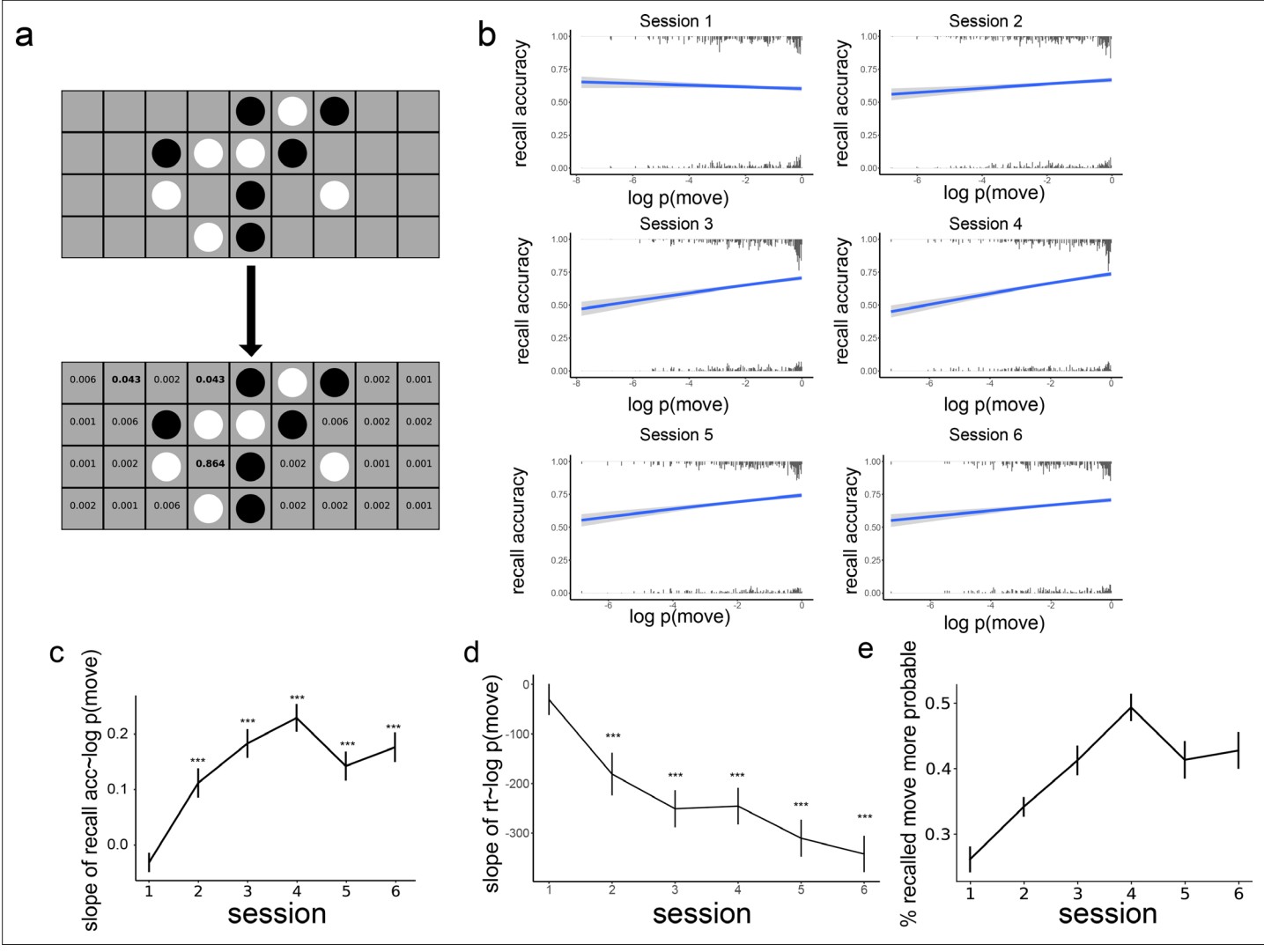

**Figure 3.** The effect of schema consistency on memory and its development across training sessions. (**a**) An example of the model's evaluation of a board where the next player is black. Based on features that would be generated by each possible next move (e.g. creating three-in-a-row), the model generates a probability distribution over potential next moves. We apply this model to the stimuli in the memory task to estimate the probability of each move. (**b**) The effect of schema consistency of a move on the recall accuracy for the move. The x-axis is the log probability of a move, which reflects schema consistency. The y-axis is the probability that a participant will remember the move. In session 1, there is no relationship between the probability of moves and subsequent memory. In sessions 2–6, people are more likely to remember schema-consistent moves. The histogram at the top and bottom of the figure is the frequency of moves with certain log probabilities that are remembered and forgotten, respectively (**c**) The relationship between move probability and recall accuracy over the 6 sessions. (*** denotes p<0.001). (**d**) The relationship between move probability and reaction time at recall for correctly remembered moves. (*** denotes p<0001). (**e**) The proportion of times the first mistake in a sequence is a more probable move than the actual move that was observed. Error bars represent the standard error of the mean, and the sample size is 35 participants.

probability of a move and the probability that the move will be remembered in session 1 ($\beta$=–0.031, $z$=–1.813, p=0.07), but in sessions 2 through 6, schema-consistent moves were more likely to be remembered (all p<0.001). This effect emerged over the first four sessions of learning (*Figure 3c*) and then dropped slightly in sessions 5 and 6.

To see whether this increase in the relationship between move probability and recall accuracy over time is significant, we aggregated the data from all the sessions and ran a mixed-effects logistic regression with subsequent memory as the outcome variable, with the probability of the move, the session the move was in, and their interactions as fixed effects, and with a participant random slope of the session and move probability. We found a main effect of session on memory ($\beta$=0.164, $z$=5.763, p<0.001) but no main effect of move probability ($\beta$=–0.0046, $z$=–0.249, p=0.804). However, there was a significant interaction between move probability and session on memory ($\beta$=0.035, $z$=7.561,

p<0.001), indicating that participants became better at remembering schema-consistent moves over the time.

In addition to recall accuracy, we looked at reaction time for placing moves at recall. We only looked at moves that were correctly recalled in each session and removed outliers with reaction times longer than 30 s (0.6% of all the correct moves were removed this way). Similar to accuracy, we found that reaction time during retrieval was initially not related to the probability of the move ($\beta$=–30.5, $t$=0.975, p=0.329) but consistently faster for more schema-consistent moves in sessions 2 through 6 (all p<0.001, *Figure 3d*). Running a mixed-effects model as for the accuracy (using linear rather than logistic regression), we found a main effect of session on reaction time ($\beta$=–156.3, $t$=–2.39, p=0.017), such that participants were faster in later training sessions. There was no main effect of move probability ($\beta$=- 46.1, $t$=–0.440, p=0.660) but a significant interaction between move probability and the session on reaction time ($\beta$=–71.06, $t$=–2.593, p=0.010), with faster responses when remembering more schema-consistent moves.

If participants are using a schema as part of their recall process, we would expect a bias toward schema-consistent moves when participants make mistakes during sequence reconstruction. To test whether this is the case, we looked at the mistakes participants made in each sequence reconstruction, measuring the fraction of the time that their answer was *more* schema-consistent than the correct answer, indicating a bias toward schema-consistent moves during recall. We only considered the first mistake, since after that mistake, the encoding and retrieval boards are different, and therefore the likely moves are no longer comparable. As can be seen in *Figure 3e*, the proportion of schema-consistent mistakes increased over the first 4 training sessions. After that, there is a drop similar to the drop in accuracy in sessions 5 and 6. We ran a mixed-effects model with a participant random slope, the proportion of schema-consistent mistakes as the outcome variable, and session as a fixed-effect predictor, and found a significant effect of the session ($\beta$=0.032, $t$=5.413, p<0.001).

Although an overall increase across sessions was observed in both schema-consistent mistakes and the relationship between recall accuracy and move probability, these effects were significantly weaker at session 5 (for schema-consistent mistakes, $t$=–2.51, p=0.02; for the relationship between recall accuracy and move probability, $t$=–3.24, p=0.001), and numerically weaker at session 6 (for schema consistent mistakes, $t$=–1.70, p=0.099; for the relationship between recall accuracy and move probability, $t$=–1.93, p=0.053). One potential explanation for this drop is that people developed new strategies in later sessions to remember schema-inconsistent moves. As can be seen in *Figure 3b*, the modeled recall accuracy for highly schema-inconsistent moves (low x values) increased from session 4–5 while the recall accuracy for schema-consistent moves remained relatively stable. This is consistent with the highest overall accuracy that occurred in session 6 (*Figure 2a*), suggesting that this decrease in slope does not come at a cost for memory accuracy. Note that reaction time did not show this weakening relationship with move probability in sessions 5 and 6, as its slope continues to decrease.

The results demonstrate that the memory benefit from gameplay training is driven in part by enhanced memory for schema-consistent moves. A possible alternative explanation of this effect is that rather than using generalized schematic knowledge, participants are in fact using episodic memories of specific move sequences that they have seen in past gameplay sessions. To test this possibility, we examined all the boards that participants saw during gameplay and the moves played on these boards, and found the repetition of exact move sequences to be very rare; out of over 1000 moves across 6 sessions there were on average 1.0 repeats (SD = 1.67) that participants experienced during gameplay and were later part of a memory sequence. Thus, the mere repetition of schema-consistent moves could not be driving the observed differences in memory. Another potential explanation could be a change in the participants' strategy. In the beginning, participants can only use their episodic memory, but over time participants could start to use a purely schema-based strategy in which they simply place moves as if they are playing the game from this board position. We simulated this strategy by drawing moves probabilistically from the gameplay (schema) model for each initial board that participants saw during the memory task in each session. We found that purely schema-based guessing can only achieve an average accuracy of 13.39% across the ix sessions (*Figure 2a*), which is much lower than participants' average recall accuracy of 65.91%. This low accuracy is due to the fact that, for most of the sequences, there are multiple plausible moves that could be played and it is therefore difficult to guess the sequence without episodic knowledge. Therefore, the improvement in performance after learning the schema is unlikely due to mere guessing of the boards.

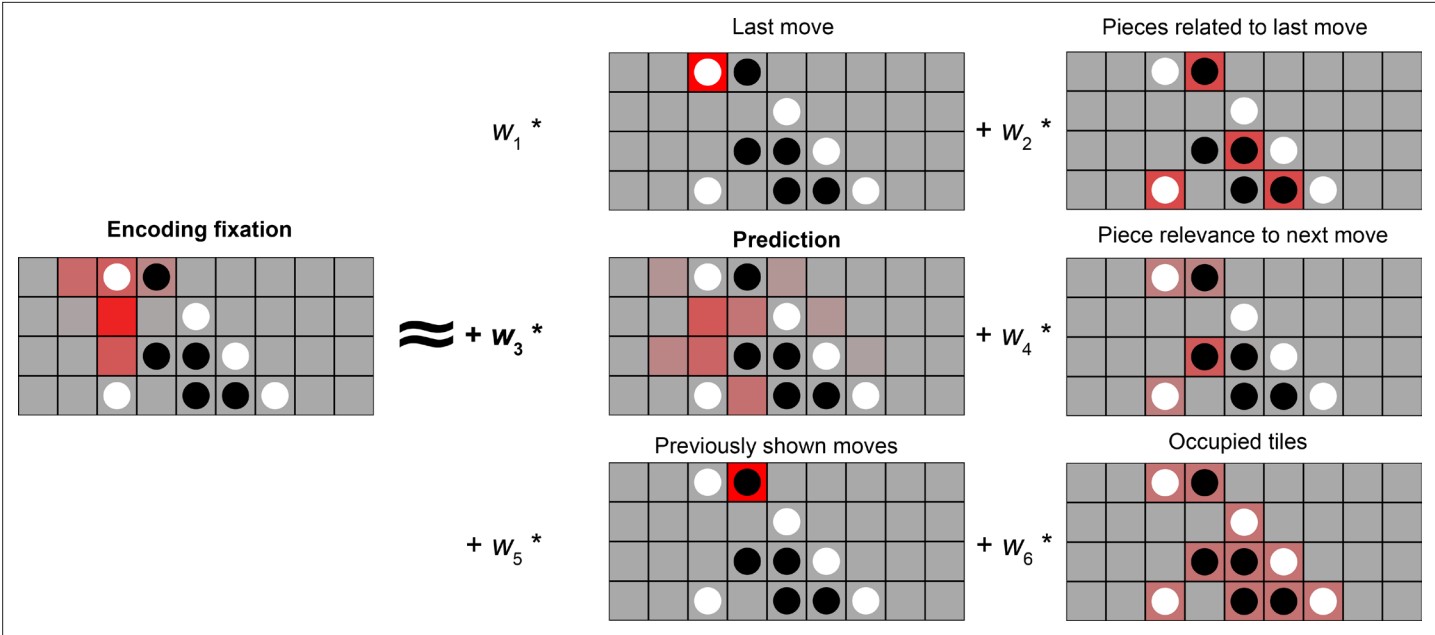

**Figure 4.** Using eye movements to reveal encoding strategies. Left: A participant's fixation heatmaps over a 5 s encoding period. right: the 6 regressors that we consider as potential encoding strategies. 1: Participants could look at the last (most recent) move, which is what they need to remember. 2: Participants could look at occupied tiles that might be relevant to the most recent move, to try to see what features the move forms. 3: Participants could be anticipating the upcoming move, meaning they will look at the empty squares on the board that are likely to be the next move. 4: Participants could also look at current pieces that are relevant for predicting the next move, i.e., pieces related to empty squares that are likely to be the next moves. 5: Participants could be looking at moves that previously appeared, in order to rehearse the observed move sequence. 6: Participants might have an overall tendency to look at occupied or unoccupied tiles.

## Eye movements became more predictive over training

A potential driver of memory improvements could be a change in encoding strategy. In particular, schemas allow people to make online predictions, which allows additional encoding time if moves are successfully predicted or generate prediction errors otherwise, both of which have been shown to be beneficial for memory (e.g. *Quent et al., 2021*). We used eye-tracking data to understand how participants' encoding strategies were related to schemas and later memory. We modeled fixations as a linear combination of six different possible strategies (*Figure 4*), including a 'prediction' strategy (regressor 3) in which fixation durations are related to the model-derived probabilities for the next move. The coefficient for each fitted regressor reflects the extent to which the strategy is used during encoding.

To look at whether the eye movements become more predictive over time, we first ran a linear mixed-effects model with prediction coefficient as the outcome variable, session as the predictor variable, and a participant random slope, to see if eye movements became more predictive over time. There was a significant fixed effect of session, $\beta=0.009$, $t=4.261$, $p<0.001$, providing evidence that people's eye movements became more predictive over training (*Figure 5a*).

## Better performance in better players is mediated by more schema-based predictions

We next sought to test whether this increase in predictive eye movements could serve as a mechanism through which expertise improves memory performance. We found that, on a session-by-session basis, more sophisticated gameplay (higher Elo) was associated with more prediction in the next memory session (Pearson $r=0.33$, $p=0.002$), demonstrating that better players predict more during encoding (*Figure 5b*). Memory sessions in which a player exhibited high levels of predictive eye movements also showed better reconstruction accuracy (Pearson $r=0.40$, $p<0.001$), despite there being no explicit demand to generate predictions during encoding (*Figure 5c*). A mediation analysis was performed to assess the mediating role of the prediction coefficient on the link between Elo and

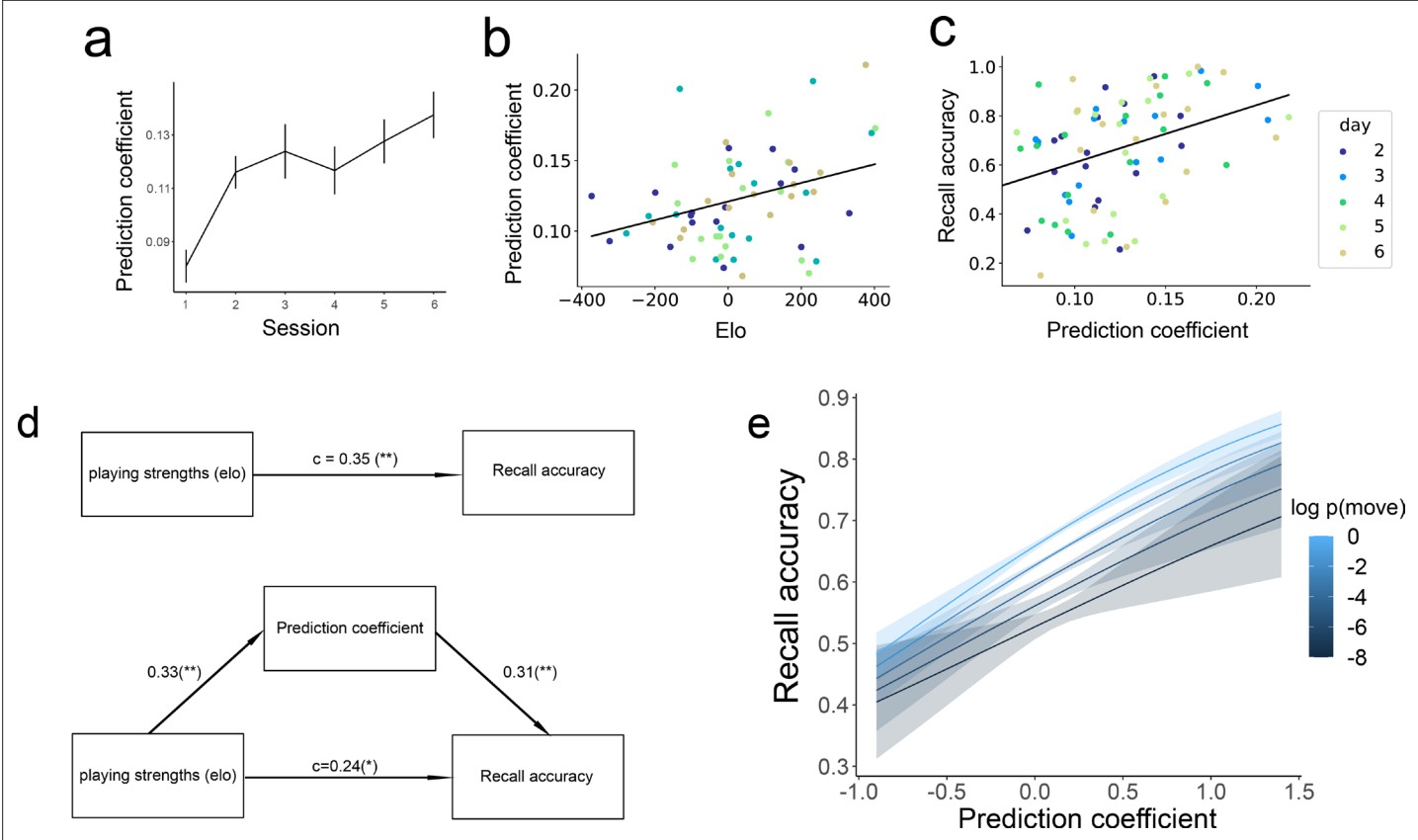

**Figure 5.** The relationship between prediction, playing strengths, and recall accuracy. (**a**) The extent to which eye movements on empty squares align with the gameplay model (prediction coefficient; see *Figure 4*) increases over training sessions. (**b**) Correlation between Elo (playing strength) and prediction coefficient in the 35 participants. (**c**) Correlation between prediction coefficient and recall accuracy. (**d**) Mediation analysis: the effect of playing strength on recall accuracy is mediated by the amount of prediction participants made during encoding. The values on the arrows represent regression coefficients for standardized measures of Elo, prediction coefficient, and recall accuracy (* denotes p<0.05, ** denotes p<0.01). (**e**) Move-level analysis on the effect of schema consistency on memory, predicting whether a move will be remembered based on the probability of the move (brighter colors are more schema-consistent) and prediction coefficient in the previous move. Error bars represent the standard error of the mean.

recall accuracy (*Figure 5d*). The total effect of Elo on recall accuracy was significant (β=0.35, t=3.26, p=0.002). The effect of Elo on the prediction coefficient was also significant, (β=0.33, t=3.13. p=0.002). The bootstrapped indirect effect was 0.10, 95% *CI* = [0.03, 0.19], p=0.006, suggesting that improved memory for better players is partially mediated by improved predictions during encoding. Significant mediation was not found for any of the other regressors (all p>0.3, except previously shown moves, p=0.056), suggesting that better prediction is uniquely important for driving the memory benefit from better gameplay. With the inclusion of the prediction coefficient, the impact of Elo on recall accuracy was reduced but remained significant (β=0.24, t=2.26, p=0.03), indicating that expertise also improves memory accuracy through additional mechanisms (at encoding and/or retrieval). We again tested whether these effects were present at the individual-participant level by running a linear mixed-effect model with participant random intercepts and slopes. Due to model convergence issues, we employed a Bayesian version of this model to regularize the coefficient estimates (see Methods). We found that the 95% credibility interval for the fixed effect of Elo on the prediction coefficient overlapped with 0 (β=0.007, 95% CI = [–0.003, 0.016]). The 95% credibility interval for the fixed effect of the prediction coefficient on recall accuracy also overlapped with 0 (β=0.164, 95% CI = [–0.54, 0.87]). Together with previous findings, these results suggest that the observed mediation effect is primarily driven by individual differences rather than within-subject development over time.

The above analysis demonstrated that making more schema-consistent predictions is correlated with better memory at the level of experimental sessions. We next constructed a logistic regression model to predict subsequent memory at the scale of individual moves, as a function of the move's

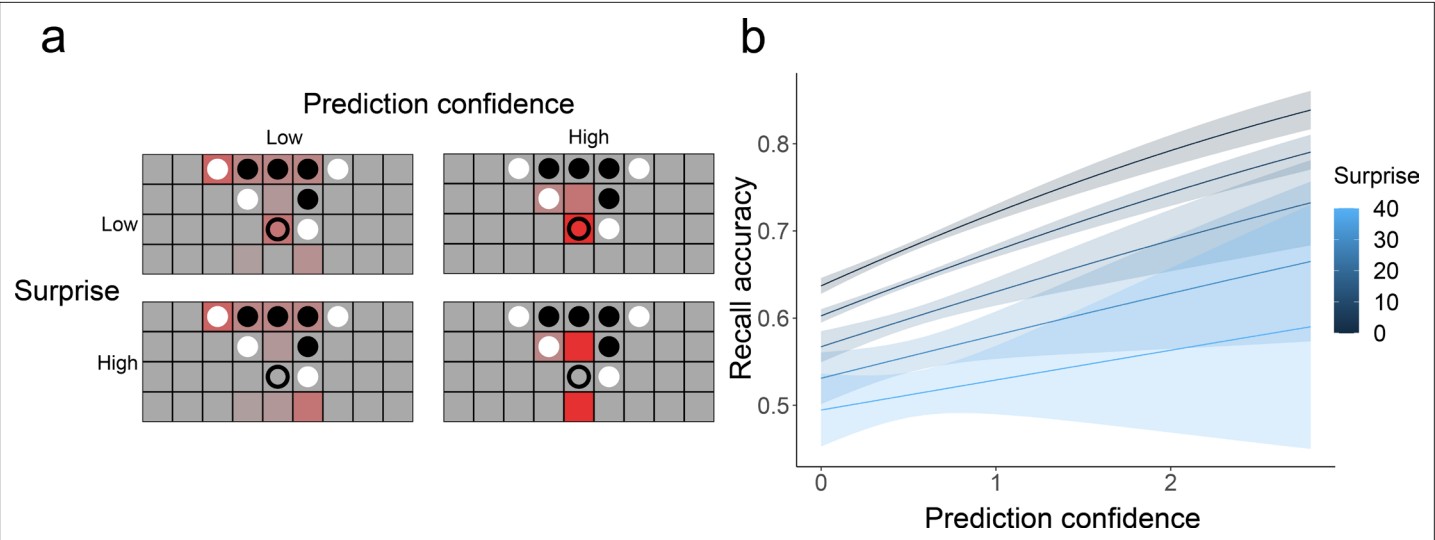

**Figure 6.** Model-free measure of prediction and their relationship with memory. (**a**) Example fixation heatmaps (in red) of high and low prediction confidence and surprise. Confidence measures the extent to which fixations were focused on a small number of unoccupied squares, while surprise measures how well these fixations aligned with the actual next move (indicated with an empty circle). (**b**) Recall accuracy was best when prediction confidence was high and surprise was low. Error bars represent the standard error of the mean.

probability and the prediction coefficient during the 5 s window before the move appeared with a subject random slope for the prediction coefficient (***Figure 5e***). There was a main effect of log probability of the move ($\beta$=0.083, $z$=5.882, p<0.001), consistent with our session-level results demonstrating a memory benefit for more likely moves. We also found a significant effect on the prediction coefficient ($\beta$=0.508, $z$=3.187, p=0.001). There is no interaction between the move probability and prediction coefficient ($\beta$=0.024, $z$=0.395, p=0.693). This result shows that making schema-consistent predictions before a move appears is correlated with better memory for the move, independent of the schema-consistency of the move.

## Model-free measures of prediction confidence and surprise

In addition to the prediction coefficient, which measured the extent to which eye movements are consistent with next-move predictions from our gameplay model, we considered two model-free aspects of eye movements on empty squares (***Figure 6a***). The first is prediction *confidence*, measuring the extent to which a participant was attending to a specific empty square (vs. looking evenly at many empty squares or at occupied squares). We use this as a measure of whether participants have a specific expectation about the next move position. The second measure is *surprise*, which is the negative log of the proportion of fixation time spent looking at the correct next move position. A high level of surprise indicates that a participant spent very little time looking at the square where the next move ended up appearing, which indicates low prediction accuracy. We then conducted a move-level logistic regression similar to the one described above, to predict whether a move will be remembered or not based on both prediction confidence and surprise while looking at the previous board (***Figure 6b***), with the subject random slope for both prediction confidence and surprise. We found the main effects of both factors, indicating that (a) making confident predictions before a move is related to better memory for that move ($\beta$=0.375, $z$=4.372, p<0.001) and (b) unexpected (surprising) moves are more poorly remembered ($\beta$=–0.015, $z$=–2.448, p=0.014).

## Discussion

The main goal of the current study was to look at how schema-based predictive processes support sequential memory encoding. Participants learned a novel schema (likely move sequences and board configurations in our-in-a-row) with much higher complexity than most artificial schemas learned in the lab. Developing a schema for moves in the game requires a significant amount of playing experience (reflected in the slow improvement in Elo ratings we observed across sessions), but can still

be accomplished within a single experiment, unlike games like chess. We observed schema-related effects on sequence memory and found that these effects emerge over time as the schema develops. We also investigated the role predictions during encoding played in the relationship between schema and memory, finding that making schema-consistent predictions, making confident predictions, and making accurate predictions were all related to successful recall at the level of individual moves.

## Schema-related memory effects and how they develop

Our behavioral findings showed that previously-studied schema effects from previous studies extend to this more complex experimental paradigm, and demonstrated how these effects developed over time. First, memory gradually improved as people's schemas gradually improved. People with a better schema had better memory, consistent with the literature on how expertise influences memory (*Chase and Simon, 1973*; *Gobet and Waters, 2003*). Second, the memory benefit over the training session was driven by the improved recall for schema-consistent information, consistent with schema literature (e.g. *Anderson, 1981*). Previous work has also shown that in some situations, schema-*inconsistent* information is better remembered (*Frank and Kafkas, 2021*), which we did not observe in this study. All the sequences we used were generated by a gameplay agent, so highly schema-inconsistent moves were rare, and none of our stimuli exhibited the extreme kind of 'contextual novelty' (*Ranganath and Rainer, 2003*; *Stark et al., 2018*) that often drives these effects; for example, we never showed a move that broke the rules of the game or played an expected sound. Future work could test whether highly schema-inconsistent items exhibit memory benefits equal to or greater than highly schema-consistent items. Third, participants' mistakes became more schema-consistent across sessions. These errors could reflect false memories for schema-consistent sequences that never occurred, similar to false memories generated by naturalistic videos or stories conforming to schematic scripts (*Lampinen et al., 2000*; *Neuschatz et al., 2002*). Alternatively, participants may be guessing schema-consistent moves when they failed to encode or retrieve the sequence; in future work, confidence responses could be used to distinguish between these possibilities. Note that some of the overall accuracy increase in the memory task across sessions could be attributable to a practice effect, but general practice-related improvement does not explain the schema-related effects that we observed on individual trials.

Interestingly, the relationship between recall accuracy and memory is attenuated in the last two sessions, driven by improved memory for schema-inconsistent moves. Expertise effects that partially extend to non-schematic situations have been previously observed; although chess experts' memory advantage was primarily driven by very high accuracy for schema-consistent (legitimate) boards in *Gobet and Waters, 2003*, experts also exhibited memory for random boards that was better than that of weaker players. One possibility is that experts may be able to devote fewer attentional resources to schematic moves, freeing up additional resources to handle more unusual moves. Participants in our study may have also developed an effortful strategy to remember schema-inconsistent moves (such as switching to a shape-based encoding for these moves), consistent with the longer reaction times observed for low-probability moves in later sessions.

There are other potential mechanisms for the schema-related memory improvement that were not explored in the current study. For example, the ways in which moves are represented in the brain may shift with schema learning, moving from a purely spatial system (e.g. remembering that a move occurred at row three, column two) to a more feature-based representation (e.g. remembered a move as 'blocking opponent's three-in-a-row'), which makes remembering easier and more robust. Future neuroimaging work can use techniques such as representational similarity analysis (*Kriegeskorte et al., 2008*) to model the representational geometry of remembered board positions and track how it changes over sessions.

## The development of predictions and their influences on memory

The eye-tracking study showed that participants spontaneously exhibited predictive eye movements consistent with the game schema, without any explicit prediction demands. This is consistent with earlier work showing automatic predictive representations in basketball experts (*Didierjean and Marmèche, 2005*). We also found that schema-based predictions were associated with improved memory (at the level of participants and the level of individual moves) and that these encoding-time predictions provide a mechanism for the impact of schema quality on memory performance. There are

several reasons why within-participant increases in Elo across sessions were not predictive of improvements in prediction and memory. First, our estimate of Elo during each memory task was based on gameplay performance on the prior day, which may not be a precise measure of the participant's current schema. Second, the biggest increase in prediction occurred between sessions one and two, but the session one memory-task data cannot be included in this prediction model since participants have not yet played the game (and we therefore cannot estimate their Elo). Third, we may simply be underpowered to detect within-participant effects, since we only collected data from six sessions for each subject. Despite the lack of within-subject effect, the session-level analysis provided the novel insight that the memory advantage for more-expert players may be driven by superior predictive processes during encoding.

Despite the numerous studies looking at how item novelty impacts memory (*Greve et al., 2017*; *Quent et al., 2021*; *van Kesteren et al., 2012*), relatively few studies have looked at the impact of temporal schematic prediction on memory for upcoming items. In a recent study, *Sherman and Turk-Browne, 2020* showed that making predictions does not benefit memory for the upcoming item and impairs the encoding of the current item, which is inconsistent with our findings. One possibility is that their study (in which participants implicitly learned arbitrary statistical dependencies between image categories presented in a continuous stream) engages a predictive process that relies on the hippocampus (*Schapiro et al., 2017*), creating interference between prediction and encoding processes. On the other hand, the explicit schematic predictions in our study that are based on rules and progression toward goal states might rely on a different network such as the medial prefrontal cortex (*Bonasia et al., 2018*; *Robin and Moscovitch, 2017*; *van Kesteren et al., 2012*).

Our eye movement data suggest that making high-confidence predictions is related to better memory, even when that prediction is inaccurate. This is in line with the generation effect (e.g. *Potts and Shanks, 2014*; *Slamecka and Graf, 1978*), which showed that people better remember unfamiliar study materials (meaning of foreign words) if they make a guess before seeing the correct answer. *Potts and Shanks, 2014* demonstrated a benefit even for minimal predictions when predictions are based on random guessing and are wrong almost all the time. In our study, although participants never made explicit predictions, high-confidence fixations (even to the wrong square) could benefit memory through a similar predictive process.

Memory is also best when predictions are most accurate (the observed move is the least surprising, coming exactly where participants anticipated it to be). Surprise or atypicality has sometimes been shown to improve memory since it attracts attention and can result in stronger encoding (*Frank and Kafkas, 2021*; *Neuschatz et al., 2002*; *Quent et al., 2021*). On the other hand, unexpected events can sometimes be difficult to retrieve at recall, since they are less connected to the schema (*Bower et al., 1979*; *Frankenstein et al., 2020*). It is possible that the degree of surprise makes a difference; as discussed above, we never presented invalid moves or unexpected categories of stimuli, which could have strongly driven attention. The type of task used might also matter, as it has been previously shown that schema-inconsistent object-location pairs are better recognized, but not better recalled when given only location as a cue (*Lew and Howe, 2017*).

Previous research has found a relationship between pupil dilation and surprise (*Antony et al., 2021*; *Lavín et al., 2014*; *Preuschoff et al., 2011*), but our study was not well-suited to measure subtle changes in pupil diameter. Studies of pupil dilation generally require tight control over the experimental stimuli, using either purely non-visual stimuli (*Preuschoff et al., 2011*), or by presenting luminance-controlled stimuli at fixation (*Lavín et al., 2014*). In our study, the luminance at fixation is highly variable since eye movements are not controlled and squares can be unoccupied (gray) or occupied by white or black pieces. Future work could attempt to study pupil dilation in this paradigm using models to control for local and global luminance (*Antony et al., 2021*) or by modifying the stimuli to ensure that all board spaces are isoluminant.

To conclude, we showed that people spontaneously engage in more sophisticated predictive processes as their schemas develop, which are beneficial for memory. The current study adds to the literature showing the adaptive values of making predictions for perception and action, and extends its benefit into the memory domain. It also provides a novel mechanism for the benefit of schemas on memory. Our paradigm uses a complex temporal schema that is much more complicated than most artificial schemas (e.g. *Tompary et al., 2020*; *van Buuren et al., 2014*) but does not take years to develop, like a chess schema (e.g. *Chase and Simon, 1973*; *Gobet and Waters, 2003*). This makes

four-in-a-row an exciting testbed for future studies of the cognitive and neural mechanisms of schema development and its impact on episodic memory.

## Methods

The study used a longitudinal design, in which participants completed six sessions over a period of about one to two weeks (*Figure 1a*). The mean interval between sessions was 2.15 days. In sessions 1–5, participants completed 2 (practice) + 30 (formal) trials of the memory task, followed by playing 40 games against an AI opponent. The gameplay task was designed to both develop participants' schema and measure their playing strength in each session. On day six, participants completed 30 trials of the memory task only. Both the memory task and the gameplay were built on Psiturk (*Gureckis et al., 2016*) and hosted on Heroku (https://www.heroku.com/).

### Participants

The first set of 19 participants (12 female, six male, and two non-binary) completed the task online on their home computers (and were instructed to maximize their browser windows during the task). The online study participants had a mean age of 20.95 years (SD = 2.84). Participants were paid $70 upon completion of the study. A second set of 16 participants (nine female, six male, and one declined to answer) performed an identical task in the lab while eye-tracking data was collected. The eye-tracking study participants had a mean age of 21.81 years (SD = 3.21). Participants were paid $100, plus up to $20 performance-based bonus. For both versions of the study, we recruited participants via online ads and personal contacts. All participants were over 18 years of age with normal or corrected-to-normal vision and gave informed consent for the study. The experimental protocol was approved by the Institutional Review Board of Columbia University (AAAS0252). One participant from the online study did not complete the last session of the study, and we have included their data for the first five sessions. All the other participants completed all six sessions of the study. Two in-person participants experienced technical issues in one of their sessions, resulting in the loss of data from one game for one participant and 15 games for the other participant. The same technical issues during the eye-tracking study resulted in a small number of trials being shown more than once to several participants; we included only data from the first presentation of each trial in the dataset.

### Experimental design

The memory task (*Figure 1b*) required participants to watch a sequence of moves and then recall the moves from memory. In each trial, participants saw an initial board for 5 s. Then one move was added to the board every 5 s. After all moves in the sequence had been added, participants completed 14 s of simple distractor math problems asking them to judge whether an equation is true or false. They had 6 s to respond to each question. After the distractor, they were shown the initial board and instructed to reconstruct the sequence in the right locations and in the right order. To account for motor mistakes, they could undo the most recent move they placed.

The first two memory trials on each day were always practice trials that had four-move sequences, and participants were given feedback on whether a move they just placed was correct or not during retrieval. Participants needed to get all moves correct in the practice trials to proceed to the formal study. The practice trials made sure participants understood the task and followed the instruction. No one was excluded from the study for failing to follow the instructions. Participants then completed 30 trials of the main memory task where they did not receive feedback during the retrieval.

The sequences of moves were generated using an AI agent from *van Opheusden et al., 2021* with an relatively weak Elo rating similar to participants' average playing strengths during their first session. We sampled from 180 unique games, each longer than 16 moves and shorter than 36 moves (i.e. the game did not end with a draw). For each session, 30 game segments with lengths ranging from four to eight moves were extracted from unique games. To ensure that meaningful schematic predictions were always possible, the first move of the sequence was always after the fifth move of the game. For example, if a game was 30 moves long and the sequence length was four, the beginning of the sequence could be anywhere between the 5th move and the 26th move of the game. Of the 30 sequences in each session, 10 sequences (two of each length) ended with one player winning (i.e. the last move created a four-in-a-row). This single set of 30 sequences for this session was shown to

all participants in a randomized order (with a different set for each session). The whole memory task takes about 50 min.

At the beginning of day one, the participants were told that the stimuli were 'circles appearing on a grid.' After the memory tasks, they were told that the stimuli were actually drawn from a game that they were about to play. They were then shown the rules of the game and played 40 games against an AI agent (*Figure 1c*), which takes about 40 min. We used a staircasing procedure, such that the agent became stronger if the participant won and weaker if the participant lost. We used the Elo rating to measure participants' playing strength (*Elo, 1978*). The ratings were computed using BayesElo (*Hunter, 2004*) to measure participants' playing strength in each session based on their performance against the AI agents. After they finished playing the games, they were asked whether they had guessed the stimuli in the memory tasks were from a game, and if so, whether they guessed the rules of the game. We did not include a control group that completed six memory sessions without being told the rules of the game or playing against the AI agent. Ensuring that participants never develop a game schema would be difficult since even without explicit instruction they could still learn the schema through implicit statistical learning or by guessing the (simple) rules of the game. Due to these challenges, we instead used the first-session performance of each participant as our no-schema control.

From Day two onwards, they were reminded of the rules of the four-in-a-row game at the beginning of the memory task. On Day six, after completing the memory task, participants completed a questionnaire asking them what strategies they used for the memory and whether their strategies changed over the course of the training.

## Gameplay model

The original model in *van Opheusden et al., 2021* used a tree search model. However, obtaining accurate move probabilities from this model would require extensive sampling. Instead, we used a feature-based myopic model of gameplay. We first defined features relevant for gameplay, which represent the relationship between the potential next move and the current board state:

1. Distance of the move from the horizontal center
2. Distance of the move from the vertical center
3. How many four-in-a-rows the move forms?
4. How many three-in-a-rows the move forms? There are three sub-categories based on the type of three-in-a-row formed: connected, disconnected, and horizontally connected that were not blocked on either side (force a win after the opponent's move, so it might have a higher value).
5. How many two-in-a-rows the move forms? There are two sub-categories based on the type of two-in-a-row formed: connected, disconnected.
6. How many opponent's three-in-a-rows the move blocks?
7. How many opponent's two-in-a-rows the move blocks? There are two sub-categories based on the type of two-in-a-row blocked: horizontally connected two-in-a-row that was not blocked in either direction (if not blocked, the opponent can force a win, so it might have a higher value), or other situations.

We then used a very strong AI agent to generate 800 games. This agent has an Elo of 365, a level similar to the best players in our study, so its move decisions can be considered an approximation of the schema of a very strong player. To fit the model, for each board state in the 800 games, we represented every possible move $x_i$ as a feature vector $F$ based on the features described above. Each move was assigned a value V that is a weighted combination of the features it forms, with weights defined by a vector $w$:

$$V\left(x_i\right) = w \cdot F\left(x_i\right)$$

We then applied a softmax function to get the probability distribution over possible moves, $p\left(x_i\right) = e^{V\left(x_i\right)} / \sum_i e^{V\left(x_i\right)}$ . We used Pytorch (*Paszke et al., 2019*) backpropagation to learn the feature weights $w$ that minimized the cross-entropy loss of the moves actually made by the agent. We trained eight models using the data from 100 of the 800 games for each model and averaged the weight vectors across models to obtain a final weight vector $w$.

We use the model's evaluation of each move as a measure of the schema consistency of a move, assuming that stronger and weaker players have qualitatively the same schema for what makes a

good move. An alternative possibility is that individuals develop schemas that are qualitatively distinct from the gameplay model, which will be reflected in some people playing moves that have a very low probability in our gameplay model. We did not find evidence for this possibility in our dataset; during the gameplay sessions, all participants tended to play moves with relatively high probability under the gameplay model (M=0.213, range=0.143–0.297, compared to a random-move baseline of 0.040). We also observed a strong correlation between Elo and this move probability measure ($r$=0.40, p<0.001), showing that stronger players are more aligned with our model. Future work (with more extensive gameplay sessions) could attempt to model unique feature weights for each specific individual.

We evaluated each move in the stimulus set with this model to determine the probability of each move under a very good strategy. Note that the stimuli were generated by a non-optimal AI agent, and, therefore, move quality varies over a wide range; the moves participants observe are often not the optimal move for a given board configuration. We also used this model to measure the probability of the move participants recalled during retrieval when they made a mistake.

## Eye-tracking

For the in-lab subset of participants, eye-tracking data was collected during the memory task on each day. The design was otherwise identical to the online experiment. Participants were seated 100 centimeters in front of a monitor and placed their heads in a chin rest 45 centimeters away from the eye-tracker. They were instructed to remain as still as possible while the eye-tracker was running and were told that they could take breaks during the experiment in between trials. Before beginning the experiment and when the participants returned from their breaks, the eye-tracker calibration and subsequent validation were done using a nine-point grid. We recorded binocular eye movement using EyeLink 1000 plus at 1000 Hz recording frequency. Light levels remained constant for the duration of the 50 min memory portion of the study. The stimuli were displayed on a 24-inch LED monitor, with a resolution of 1920 by 1080 pixels and a refresh rate of 60 Hz. The outputted EDF files were converted to asc files and parsed with PyGaze (*Dalmaijer et al., 2014*).

Fixation maps were created for each 5 s period during which an initial board was shown or a move was shown. To handle uncertainty in assigning gaze to squares, we performed a soft assignment to board locations based on distance. For a fixation at the position $x_F$ with duration $t_F$ the square with center coordinate $x_i$ was assigned a fixation weight of

$$t_F \cdot \frac{e^{-\|x_F - x_i\|_2 /25}}{\sum_j e^{-\|x_F - x_j\|_2 /25}}$$

Here, distance is in the unit of pixels. Since the length of the square is 136 pixels, the smoothing temperature of 25 is approximately 1/5 the length of a square and, therefore, is only relevant for fixations close to square boundaries. The weights for all fixations during the 5 s window were summed to obtain a final map of fixation weights for all board squares.

## Eye movement regression model

We modeled fixation maps as a linear combination of six potential strategies that could be used during memory encoding. For each board, the regressors, described below, are length-36 vectors that correspond to the 36 tiles in the game.

1. The most recent move: has a value of one in the square that corresponds to the most recent move, and 0 s elsewhere (all 0 s at the initial board). This regressor captures the extent to which participants are looking at the move that just appeared on the board.
2. Pieces related to the most recent move: has a value of one in occupied squares that are related to the most recent move (within three tiles from the most recent move in any direction), and 0 s elsewhere. This regressor captures the behavior of looking at how a move relates to previously-placed pieces (e.g. to detect whether it adds onto a line of same-color pieces).
3. Anticipation of the upcoming move (prediction): for the unoccupied squares, is equal to the probabilities of that square being the location of the next move, as calculated by the near-optimal gameplay model. For occupied squares, has a constant value equal to the mean value of the unoccupied squares. This regressor measures how well participants' eye movements predict likely positions for the next move.

4. Piece relevance to the next move: for each occupied square, is the sum of the next-move probabilities for all empty squares that are related to it (within three squares). For unoccupied squares, its value is equal to the mean value of the occupied squares. This regressor reflects the tendency to focus on squares that are likely to be related to the next move.
5. Observed previous moves: has a value of one for the moves seen earlier in the sequence (excluding the most recent move), 0 s elsewhere. This reflects a strategy of reviewing previous moves during memory encoding.
6. Occupied tiles: has values of 1 s on occupied tiles and 0 s on unoccupied tiles. This measures the preference for looking at occupied squares versus empty squares.

Each regressor and the eye movement fixation maps were z-scored within each board. We concatenated each of these across all the boards that each participant saw during a session and ran a multiple linear regression (with the fixation maps as the outcome variable, and the six regressors as the predictor variables). The coefficient for each regressor reflects the extent to which this participant used this eye movement strategy when encoding boards in this session. We also ran this regression separately for each individual board (for each move shown during encoding) to obtain trial-level estimates of eye movement strategy.

## Regression models

For each model, we started with the most complex frequentist model, including a subject slope for all of the predictors. In case they did not converge, we used simpler models with either fewer random slopes or just a random intercept. If this happens, we also checked whether the effect holds with a more complex model using a Bayesian model. The reported effects with more simpler frequentist models have been shown to hold with a more complex Bayesian model. If a different result was obtained, we report the results of the more complex Bayesian model. For frequentist models, we use lme4 pacakge (*Bates et al., 2015*). For the Bayesian models, we used the default settings of rstanarm package (*Goodrich et al., 2022*).

## Mediation analysis

The total effect was calculated by running a linear regression predicting recall accuracy from Elo. Next, we calculated the effect of Elo on the prediction coefficient and the effect of Elo and the prediction coefficient on recall accuracy. The significance of the mediation was computed with the package *mediation* (*Tingley et al., 2014*) that used a bootstrapping procedure. Standardized indirect effects were computed for each of the 10,000 bootstrapped samples, and the 95% CI was computed by determining the indirect effects at the 2.5th and 97.5th percentiles.

## Estimating trial-level prediction confidence and surprise

We constructed two trial-level measures of fixation statistics which allowed us to describe eye movement strategies at a finer scale and in a model-free way (without assuming that participants were making predictions according to our schema model). The first is prediction confidence, which measured the extent to which a participant spent time focused on specific empty squares. High values of confidence indicate that a participant spent a large fraction of the trial looking at only a small number of empty squares, indicating a strong prediction about the upcoming move. We compute this as the expected information gain between a uniform distribution over all empty squares and the fixation distribution. Given the fixation time $T(x_i)$ for each square $x_i$, we define $P(empty)$ as the fraction of the 5 s window spent fixating on empty squares, and $P(x_i) = T(x_i)/P(empty)$ as the normalized fixation distribution over empty squares. The information gained from a fixation is 0 for fixations on occupied squares, and for fixations on empty squares reflects the entropy difference between a uniform distribution and the fixation distribution. Therefore, we define:

$$\text{Prediction confidence} = P(empty) \cdot \left( log(N_{empty}) - \sum_i^N P(x_i) \, logP(x_i) \right)$$

The second is prediction surprise, indicating the extent to which a participant failed to look at the location where the next move was going to appear. High values of surprise indicate that the move appeared in a location that the participant spent very little time looking at. This is defined as the negative log of the percentage of time participants looked at the correct upcoming move position $x_{next}$:

$$\text{Prediction surprise} = -log\left(P\left(x_{next}\right)\right)$$

## Additional information

### Funding

| Funder | Grant reference number | Author |
|---|---|---|
| Columbia University | Graduate Student Fellowship | Jiawen Huang |
| Columbia University | start-up funding | Christopher Baldassano |
| National Institutes of Health | R21MH126269 | Wei Ji Ma |
| National Institutes of Health | R01MH118925 | Wei Ji Ma |
| National Science Foundation | 2008331 | Wei Ji Ma |

The funders had no role in study design, data collection and interpretation, or the decision to submit the work for publication.

### Author contributions

Jiawen Huang, Conceptualization, Data curation, Software, Formal analysis, Investigation, Visualization, Methodology, Writing - original draft, Project administration, Writing - review and editing; Isabel Velarde, Data curation, Project administration, Writing - review and editing; Wei Ji Ma, Conceptualization, Resources, Software, Supervision, Methodology, Writing - review and editing; Christopher Baldassano, Conceptualization, Resources, Software, Formal analysis, Supervision, Funding acquisition, Validation, Visualization, Methodology, Project administration, Writing - review and editing

### Author ORCIDs

Jiawen Huang http://orcid.org/0000-0003-1362-0412
Isabel Velarde http://orcid.org/0000-0001-5639-0907
Wei Ji Ma http://orcid.org/0000-0002-9835-9083
Christopher Baldassano http://orcid.org/0000-0003-3540-5019

### Ethics

The experimental protocol was approved by the Institutional Review Board of Columbia University. (AAAS0252) All participants were over 18 years of age with normal or corrected-to-normal vision, and gave informed consent.

### Decision letter and Author response

Decision letter https://doi.org/10.7554/eLife.82599.sa1
Author response https://doi.org/10.7554/eLife.82599.sa2

## Additional files

### Supplementary files
• MDAR checklist

### Data availability
All the data is openly available through https://osf.io/29cpg/.

The following dataset was generated:

| Author(s) | Year | Dataset title | Dataset URL | Database and Identifier |
|---|---|---|---|---|
| Jiawen H, Isabel V, Wei JM, Christopher B | 2022 | Schema-based predictive eye movements support sequential memory encoding | https://osf.io/29cpg/ | Open Science Framework, 10.17605/OSF.IO/29CPG |

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
