## [Editor Report]

This work designed a novel board game paradigm (4-in-a-row, a sort of tic-tack-toe expansion) in combination with eye movement recordings to examine how schemas gradually learned in the game influence memory. They provide impressive evidence that, in both behavior and eye movements, schema indeed guides the encoding of sequential events and facilitates memory performance. This paper would be of great interest to many fields, such as memory, learning, and decision-making.

---

## [Decision Letter]

**Decision letter after peer review:**

Thank you for submitting your article "Schema-based predictive eye movements support sequential memory encoding" for consideration by *eLife*. Your article has been reviewed by 3 peer reviewers, one of whom is a member of our Board of Reviewing Editors, and the evaluation has been overseen by Timothy Behrens as the Senior Editor. The following individual involved in review of your submission has agreed to reveal their identity: Zachariah M. Reagh (Reviewer #2).

The reviewers have discussed their reviews with one another, and the Reviewing Editor has drafted this to help you prepare a revised submission. All the reviewers agree that it is an impressive and well-done study to explore a very important question, i.e., the nature and origins of schema-based effect on memory. The clever experimental paradigm strikes a nice balance between complexity in capturing elements of naturalistic human behavior and simplicity in terms of analyses. Meanwhile, the reviewers also raised several critical concerns that need further control experiments and analyses to support the conclusion.

Essential revisions:

1) Instead of memorizing the board positions, subjects might simply consider how they play from that position, a strategy that could account for all the observations. To decouple the schema-based and purely-memory-based influences, a further control study needs to be conducted (see details in Reviewer 3's comments);

2) There lacks a control group who neither knows the rule nor play the game before. The prediction is that the control group will show (1) worse memory; (2) their error patterns are different and unrelated to p (move);

3) Reanalyzing the data to address the potential mixture of session-level and participant-level variability (see Reviewer 1's 2nd concern)；

4) Further clarifications are needed, such as the drop-off in late sessions, reasons for not looking at pupil dilation, the rationale for schema-based memory facilitation hypothesis, and presentation of RT data (see Reviewer 2's comments)

*Reviewer #1 (Recommendations for the authors):*

This paper develops an interesting and new experimental paradigm to investigate how schema participates in memory and helps to generate predictions, using behavioral measurements, eye movement recordings, and computational modeling. The results show the correlation between memory accuracy, game performance, and predictive eye movements, supporting that schema significantly boosts memory and guide prediction. This paper would be of potential interest to many fields, such as memory, learning, and decision-making.

We see two major strengths in the paper. First, although schema-based memory is crucial, it is very hard to study it in a natural and also controllable way in the lab. This work develops a complex game paradigm with proper difficulty levels, enabling the examination of the characteristics of the process. Second, the combination of memory behavior, eye movement, and game playing provides a comprehensive framework to tackle the question.

The major weakness of the work mainly lies in its lack of a control group, that is, subjects who neither know the rule nor play the game before. This would be important evidence to support the advantage of schema-based memory boosting. Moreover, the source of the effects needs to be further clarified using additional analyses.

Concerns:

1. Schema-based boost in memory and predictive eye movement lacks a control group who neither know the rule nor play the game before. The prediction is that the control group will show (1) worse memory; (2) their error patterns are different and unrelated to p (move).

2. Most of the correlation analyses mixed the variability levels and thus confused the source of the memory boosting effects. For example, when calculating the correlation results (Figure 2C), each data point represents a session of a participant. In this case, the source of the effect could come from either session-level variability that is due to training or participant-level variability. In fact, all the linear mixed-effect models have the same problem and need to be fixed. I would not suggest excluding the random effect for slopes since that would increase the probability of Type I error and as a consequence, the observed effect might arise from individual differences rather than experimental manipulation (see Brauer and Curtin, Psychol Methods, 2018 and Bates, arXiv, 2018 for guides). It is important to include individual-level random effects in the analysis.

*Reviewer #2 (Recommendations for the authors):*

This paper by Huang and colleagues uses a simple board game, "4-in-a-row," to test the relationship between schemas and episodic memory. Specifically, the goal here was to examine the way schemas can guide encoding of sequential events as the game unfolded. In sum, the authors report that memory for move sequences in the game improved with player skill, and that memory was particularly improved for schema-consistent sequences. Further, eye movements during encoding were associated with stronger memory for move sequences, which was most prevalent in expert players.

We think this study does a fine job of answering how the development of schema aid in prediction and thereby memory through use of a stimulus that is at once naturalistic but learnable by participants in laboratory time. This is a difficult set of boxes to check. While 4-in-a-row is certainly a simplistic way of testing encoding and retrieval of "events," it nonetheless tests memory in a fairly rich and meaningful way compared to much existing research, especially with regard to sequential order. The approach itself strikes a really nice balance between complexity in capturing elements of naturalistic human behavior, and simplicity in terms of analyses. Overall we think this experiment is sound, and the paper is well-written.

We have some requests for expansion/clarification and a few suggestions for strengthening the paper.

1. Looking at Figure 3c and 3d, it struck us that the slope of recall accuracy increases linearly across sessions 1-4, but then drops pretty markedly to sessions 5 and 6. If things simply leveled-off or dropped slightly, it would seem less important or interesting, but the drop-off is quite stark. What could be happening here? What do the authors make of this?

2. It was actually a bit surprising to us that the authors didn't use pupillometry when evaluating surprise. Pupil dilation has been strongly linked to the constructs of surprise and prediction error, more so than gaze. We are not necessarily suggesting that the authors rerun their model using pupillometry, but some rationale for why gaze rather than pupil dilation was optimal for testing surprise in this context would help. As it stands, we felt that this disconnect left the reasoning behind the surprise model feeling unclear, and we think a more explicit discussion of this would be helpful.

3. As the authors themselves flesh out in the Discussion, there is a fair amount of inconsistency in the literature as to whether schema-consistent (or "congruent," as others often say) or schema-inconsistent information provides relative a boost to episodic memory. And although we think their points related to this in the Discussion are fine, we think it would be worth laying out the reasoning for the prediction that schema-consistency should benefit memory in the Introduction. Our reasoning here is that, from our point of view, it could have easily gone either way. And we think that the field can benefit at this point from clear logic laying out predictions related to schema and memory.

4. Did the authors analyze reaction time data at all? To be clear, we are not making a strong suggestion to do so, but given links between surprise and response time variability, and the logical relationship between increasing experience and time taken to do something, it seems a logical thing to do in characterizing performance.

*Reviewer #3 (Recommendations for the authors):*

This paper presents a series of analyses of the relationship between predictive schemas – representations that allow people to form predictions about events in their environment – and performance in a memory task. Participants in the experiments learned to play a novel board game over a period of several days, and also performed a memory task in which they had to recall sequences of board positions. The first time they performed the memory task they had no experience with the game, so these just looked like random configurations of black and white circles. As players became more familiar with the game, the memory performance also improved.

Several analyses dig deeper into this phenomenon, examining how participants' memory for specific board positions are related to the probability of the corresponding moves in the game, as estimated by an AI agent. This analysis shows that people have better memory for higher probability moves. By bringing another group of participants into the laboratory and performing eye tracking, it was shown that people's eyes movements also seem to be influenced by the probability of moves. Further analyses that express confidence and surprise in terms of these eye movements – how focused people seem to be on a specific position and how long they spend looking at the correct position – show that these quantities are also related to memory.

Overall, I think this is an impressive piece of work that does a nice job of exploring several questions about the nature and origins of schema-based effects on memory performance through the clever use of a novel task. However, I do think there is one substantial potential confound. As people are getting better at playing the game, it seems like one potential strategy they might use is to worry less about memorizing the board positions and instead to simply consider how they would play from that position. People following this strategy would seem to account for all of the results that are presented: it would result in a higher probability of being correct on higher probability moves, and potentially produce the associated effects involving eye-movements since those eye-movements tend to be concentrated on the high probability moves as well.

This kind of confound is likely to arise in studies looking at schema effects on memory, because if people have a schema they can use that schema to make informed guesses that are more likely to be right for schema-consistent items. However, in this setting it seems like there is actually an opportunity to address this confound directly. I think this could be done by conducting a further study in which people play from the same board positions that are used to initiate the memory recall sequences – perhaps doing so on different days so as to remove concerns about interference. By measuring the actual play sequences that people generate, it would be possible to focus the analyses on cases where the memory sequences differ from these play sequences. In fact, if the play sequences are generated the day before the memory test, the memory test could include two kinds of trials: one set where the memory sequence corresponds to that produced by the participant on the previous day, and one where it diverges. If the reported effects are observed in both of these conditions, that helps to decouple simulated play from generic schema use (that is, if a move in a memory sequence that has high probability but wasn't played by the participant the previous day is still remembered well, it provides strong support for a schema effect rather than a play-based strategy).

[Editors' note: further revisions were suggested prior to acceptance, as described below.]

Thank you for resubmitting your work entitled "Schema-based predictive eye movements support sequential memory encoding" for further consideration by *eLife*. Your revised article has been evaluated by Timothy Behrens (Senior Editor) and a Reviewing Editor.

Since the original two reviewers were not available to review the revised version, we have invited Reviewer 4 to evaluate the revision.

The manuscript has been improved but there are some remaining issues that need to be addressed, as outlined below:

1) Providing the learning curve of the control group (first session here) instead of the averaged results;

2) Please see detailed comments by Reviewer 4.

*Reviewer #1 (Recommendations for the authors):*

About the two major comments or experiments I raised, the authors have done a fine job addressing them. They have performed a simulation analysis to exclude the alternative explanation raised in the first question. They chose to use the first session of the data as the control group to answer the second question.

Their answers are fine but I would like to clarify my original point that may have been unclear. That is, showing the learning curve rather than a single point of the control group's performance, i.e., the first session, as a control.

*Reviewer #4 (Recommendations for the authors):*

Huang et al. examined the influence of schemas on prediction and sequence memory for a novel board game. Participants' memory for sequences of moves was tested across several sessions as they learned the game and improved. Over the sessions, and accompanied by a skill increase, participants' memory for the sequences improved. This learning was also related to eye-movements during encoding which were predictive of memory for the sequences.

This study does an excellent job of linking schema development to prediction/attention and memory.

Comments:

In the first paragraph of the Results section, it was unclear whether during the game play task participants were provided instructions or whether they had to learn the game through some kind of trial and error. It should be specified that they were provided with rules since early in the paragraph it notes that the participants were naive to the fact that a game was being shown in the initial memory test phase.

"Modeling schema-related memory improvement" In this section a model is trained based on moves from a near optimal AI. A probability distribution is made for upcoming moves based on the likelihood a move would be made by a near optimal AI. This probability is interpreted to be related to the schema consistency of a move. This may be true for an expert player. However, a near optimal move may not be schema-consistent for a new player or maybe even an average player. The participant's internal schema when they are beginners and progress may not map on to the near optimal AI moves. This is an interesting analysis, but calling the near optimal AI move schema consistent just slightly seems like a miswording for me.

Discussion "One possibility is that their study (in which participants implicitly learned statistical dependencies between image categories presented in a continuous stream) engages a predictive process that relies on the hippocampus (Schapiro et al., 2017), creating interference between prediction and encoding processes, while the explicit schematic predictions in our study could rely on a different network such as the medial prefrontal cortex " While this is a compelling assertion, can you describe anything that psychologically varies between these tasks? For example, Shapiro et al. had participants learn arbitrary statistical sequences, but here the sequences could be understood in the context of rules, goals, and progression towards goal states.

Methods: During the memory task, 10 out of the 30 sequences ended in a win state. Since a win-state caps off a sequence this could be thought of as an event boundary. Was there any effect of these 10 sequences that differed from the other 20 or interacted with other effects? For example, if a win-state is an event boundary, is particularly salient, and is better remembered, then a participant merely has to reconstruct the sequence from probe to win-state, rather than remembering each move.

Modeling schema-related memory improvement" In this section the authors missed a period in "near- optimal AI agent The model identifies".

---

## [Author Response]

Essential revisions:1) Instead of memorizing the board positions, subjects might simply consider how they play from that position, a strategy that could account for all the observations. To decouple the schema-based and purely-memory-based influences, a further control study needs to be conducted (see details in Reviewer 3's comments);

We have now conducted an additional analysis, in which we simulate schema-consistent guessing on the memory task using our AI game-playing model. Because this model was trained on gameplay distribution similar to those used in the memory task, this provides an upper bound on the best-possible performance achievable without using any episodic memory. We included a discussion of this simulation in the result section:

“Another potential explanation could be a change in participants’ strategy. In the beginning, participants can only use their episodic memory, but over time participants could start to use a purely schema-based strategy in which they simply place moves as if they are playing the game from this board position. We simulated this strategy by drawing moves probabilistically from the gameplay (schema) model for each initial board that participants saw during the memory task in each session. We found that purely schema-based guessing can only achieve an average accuracy of 13.39% across the 6 sessions (Figure 2a), which is much lower than participants’ average recall accuracy of 65.91%. This low accuracy is due to the fact that, for most of the sequences, there are multiple plausible moves that could be played and it is therefore difficult to guess the sequence without episodic knowledge. Therefore, the improvement in performance after learning the schema is unlikely due to mere guessing of the boards.” (line 268-280)

This simulation demonstrates that, even under optimal guessing, it is not possible for a non-episodic strategy to produce our observed results. We therefore do not believe it is necessary to run the proposed behavioral experiment with human participants, since selecting moves based on how the participant would play the game cannot yield performance higher than 13.39%.

Additionally, if participants were no longer attempting to memorize the specific moves being played, the time spent looking at the most recent move on the board would drop sharply after participants developed a schema. We therefore looked at the change across sessions for the last-move coefficient in our eye-movement analysis, which describes the extent to which participants look at the most recently-placed move. Although the last move coefficient did reduce numerically over time, this reduction is small and not significant (p = .114).

**Author response image 1. sa2fig1:** 

Based on the analyses presented above, we believe that a purely schema-based strategy is unlikely to explain the effects we observed in the study.

2) There lacks a control group who neither knows the rule nor play the game before. The prediction is that the control group will show (1) worse memory; (2) their error patterns are different and unrelated to p (move);

The first session for each participant did serve as exactly this control group, which we have now emphasized in the beginning of the result section.

“In the first session, participants were not told that these stimuli came from a game, and were only instructed to remember circles appearing on a grid. The first session therefore provided a no-schema baseline, since participants could not use a game model to make predictions about upcoming moves. In a post-task questionnaire, we confirmed that in session 1 most participants did not suspect that the stimuli were from a game or guess the rule of the game (14 out of 19 in the online study, and 13 out of 16 in the in-person study).” (line 111-114)

We found that subjects indeed had worse memory in this first session (figure 1) and that their recall accuracy was not related to the probability of the move (figure 3b, top left figure).

3) Reanalyzing the data to address the potential mixture of session-level and participant-level variability (see Reviewer 1's 2nd concern);

We now include random participant intercepts and slopes for *all* our regression models in the study, following the approach of *Brauer and Curtin, Psychol Methods, 2018* (as recommended by Reviewer #1). In cases where this model failed to converge, we *simplify the model,* and used Bayesian priors to stabilize the coefficient estimates. This approach yielded conclusions largely similar to our previous (random-intercept-only) models for almost all analyses (lines 126-128, 171, 180-181, 197, 223, 295-296, 352-356, 372-377).

We described our approach for using frequentist and Bayesian models in the Methods:

“Regression Models

For each model, we started with the most complex frequentist model, including a subject slope and intercept for all the predictors. In cases where this model did not converge, we took two parallel approaches. First, we simplified the frequentist model by removing random slopes for some or all of the effects until the model converged. We also fit a Bayesian version of the complex model (with all random slopes). If these two approaches yielded the same result (with the same coefficients statistically different from zero), we reported the simpler frequentist model; if the results differed, we reported the result for the full Bayesian model. For frequentist models, we use lme4 pacakge (Bates et al., 2015). For the Bayesian models, we used the default settings of rstanarm package (Goodrich et al., 2022).” (line 719-727)

Accounting for across-subject variability did yield a different outcome for our analyses examining the relationship between Elo, prediction coefficient, and memory accuracy:

“To understand whether this relationship was present within individual participants, we fit a linear mixed effect model to predict memory performance from Elo with per-participant intercepts and slopes as random effects. We found that the relationship between Elo and memory accuracy was not significant in this model (β = 0.012, t = 1.196, p = .242), suggesting that this effect was primarily driven by individual differences (people with better schema tend to have better memory) rather than across-session improvements in Elo.” (line 143-148)

“We again tested whether these effects were present at the individual-participant level by running a linear mixed effect model with participant random intercepts and slopes. Due to model convergence issues, we employed a Bayesian version of this model to regularize the coefficient estimates (see Methods). We found that the 95% credibility interval for the fixed effect of Elo on prediction coefficient overlapped with 0 (β = 0.007, 95% CI = [-0.003, 0.016]). The 95% credibility interval for the fixed effect of prediction coefficient on recall accuracy also overlapped with 0 (β = 0.164, 95% CI = [-0.54, 0.87]). Together with previous findings, these results suggest that the observed mediation effect is primarily driven by individual differences rather than within-subject development over time.” (line 317-328)

We included in the discussion some interpretation of this results.

“There are several reasons why within-participant increases in Elo across sessions were not predictive of improvements in prediction and memory. First, our estimate of Elo during each memory task was based on gameplay performance on the prior day, which may not be a precise measure of the participant’s current schema. Second, the biggest increase in prediction occurred between sessions one and two, but the session one memory-task data cannot be included in this prediction model since participants have not yet played the game (and we therefore cannot estimate their Elo). Third, we may simply be underpowered to detect within-participant effects, since we only collected data from six sessions for each subject. Despite the lack of within-subject effect, the session-level analysis provided the novel insight that the memory advantage for more-expert players may be driven by superior predictive processes during encoding.” (line 419-429)

4) Further clarifications are needed, such as the drop-off in late sessions, reasons for not looking at pupil dilation, the rationale for schema-based memory facilitation hypothesis, and presentation of RT data (see Reviewer 2's comments)

Presentation of RT data

Thank you for the suggestion to include the RT data. It indeed showed a similar pattern to what we observed in accuracy data, further strengthening our conclusions about the relationship between schema-consistency and memory. We’ve included a discussion about this in the results:

“In addition to recall accuracy, we looked at reaction time for placing moves at recall. We only looked at moves that were correctly recalled in each session and removed outliers with reaction times longer than 30 sec (0.6% of all the correct moves were removed this way). Similar to accuracy, we found that reaction time during retrieval was initially not related to the probability of the move (*β* = -30.5, *t* = 0.975, *p* = .329) but consistently faster for more schema-consistent moves in session 2 through 6 (all *p* <.001, figure 3d). Running a mixed-effects model as for the accuracy (using linear rather than logistic regression), we found a main effect of session on reaction time (*β* = -156.3, *t* = -2.39, *p* = .017), such that participants were faster in later training sessions. There was no main effect of move probability (*β* = – 46.1, *t* = -0.440, *p* = .660) but a significant interaction between move probability and session on reaction time (*β* = – 71.06, *t* = -2.593, *p* = .010), with faster responses when remembering more schema-consistent moves.” (line 200-210)

We also modified figure 3 to include the figure showing the change in slope.

Drop-off in late sessions:

The reviewers are correct that, in the late sessions, move probability is less strongly related to memory accuracy and memory errors. We now statistically quantify this drop-off and discuss how participants may be changing their memory strategies in the final sessions of the experiment in order to further improve their accuracy:

“Although an overall increase across sessions was observed in both schema-consistent mistakes and the relationship between recall accuracy and move probability, these effects were significantly weaker at session 5 (for schema-consistent mistakes, *t* = -2.51, *p* = .02; for the relationship between recall accuracy and move probability, *t* = -3.24, *p* = .001), and numerically weaker at session 6 (for schema consistent mistakes, *t* = -1.70, *p* = 0.099; for the relationship between recall accuracy and move probability, *t* = -1.93, *p* = .053). One potential explanation for this drop is that people developed new strategies in later sessions to remember schema-inconsistent moves. As can be seen in figure 3b, the modeled recall accuracy for highly schema-inconsistent moves (low x values) increased from session 4 to 5 while the recall accuracy for schema-consistent moves remained relatively stable. This is consistent with the highest overall accuracy occurred in session 6 (figure 2a), suggesting that this decrease in slope does not come at a cost for memory accuracy. Note that reaction time did not show this weakening relationship with move probability in session 5 and 6, as its slope continues to decrease.” (line 225-237)

We also included a paragraph in the discussion about possible explanations for the observed effects.

“Interestingly, the relationship between recall accuracy and memory is attenuated in the last 2 sessions, driven by improved memory for schema-inconsistent moves. Expertise effects that partially extend to non-schematic situations have been previously observed; although chess experts’ memory advantage was primarily driven by very high accuracy for schema-consistent (legitimate) boards in Gobet and Waters (2003), experts also exhibited memory for random boards that was better than that of weaker players. One possibility is that experts may be able to devote fewer attentional resources to schematic moves, freeing up additional resources to handle more unusual moves. Participants in our study may have also developed an effortful strategy to remember schema-inconsistent moves (such as switching to a shape-based encoding for these moves), consistent with the longer reaction times observed for low-probability moves in later sessions.” (line 435-445)

The rationale for schema-based memory facilitation hypothesis

We modified the introduction to provide more justification for the hypothesis of schema-consistent memory benefits.

“Although previous research has sometimes found novelty-driven memory improvements for schema-inconsistent information (reviewed in Frank and Kafkas, 2021), studies of expert memory for complex memoranda such as chess boards have shown an advantage for schema-consistent stimuli (e.g., board positions from actual chess games) (Gobet and Waters, 2003). Thus, we hypothesized that the memory improvement resulting from the development of schema in 4-in-a-row should similarly be specific to moves that are schema-consistent.” (line 82-88)

Reasons for not looking at pupil dilation

The study was not designed to measure subtle changes in pupil diameter, and we have now added a paragraph about this limitation to the Discussion:

“Previous research has found a relationship between pupil dilation and surprise (e.g., Antony et al., 2021; Lavín et al., 2014; Preuschoff et al., 2011), but our study was not well-suited to measure subtle changes in pupil diameter. Studies of pupil dilation generally require tight control over the experimental stimuli, using either purely non-visual stimuli (Preuschoff et al., 2011), or by presenting luminance-controlled stimuli at fixation (Lavín et al., 2014). In our study the luminance at fixation is highly variable, since eye movements are not controlled and squares can be unoccupied (grey) or occupied by white or black pieces. Future work could attempt to study pupil dilation in this paradigm using models to control for local and global luminance (Antony et al. 2021) or by modifying the stimuli to ensure that all board spaces are isoluminant.” (Line 508-516)

That said, we did look at the time course of pupil size over time (5 sec before a move shows up and 5 sec after, downsampled to 100 Hz) when surprise (as defined by the extent to which pre-move fixations were in the incorrect location) is high (top 25%), low (bottom 25%), and medium (50% in the middle). We found that there was a consistent pattern of pupil size across trials, a ramp in size during each trial peaking about 300 ms after the presentation of the next move, and followed by a sharp constriction. This kind of decrease at event switches has also been observed at event boundaries in movies, and is hypothesized to be due to a drop in cognitive load at boundaries (Smith et al., 2006). As shown in Author response image 2, high-surprise moves did indeed have larger pupil sizes just after the move (at 300ms) and after the constriction (at 500-800ms). However, this effect is present even before the move is shown (starting at -4000ms), suggesting that surprise may be related to pre-move attentional state or may be more likely to occur on certain kinds of boards (e.g. boards with many pieces). Although this is an interesting result, we believe that it is out of the scope for the current paper. We are conducting a follow-up study in which the move participants see is independently manipulated to have different levels of surprise, and hope to more comprehensively explore the question about pupil size and surprise in this paradigm.

Reviewer #1 (Recommendations for the authors):This paper develops an interesting and new experimental paradigm to investigate how schema participates in memory and helps to generate predictions, using behavioral measurements, eye movement recordings, and computational modeling. The results show the correlation between memory accuracy, game performance, and predictive eye movements, supporting that schema significantly boosts memory and guide prediction. This paper would be of potential interest to many fields, such as memory, learning, and decision-making.We see two major strengths in the paper. First, although schema-based memory is crucial, it is very hard to study it in a natural and also controllable way in the lab. This work develops a complex game paradigm with proper difficulty levels, enabling the examination of the characteristics of the process. Second, the combination of memory behavior, eye movement, and game playing provides a comprehensive framework to tackle the question.The major weakness of the work mainly lies in its lack of a control group, that is, subjects who neither know the rule nor play the game before. This would be important evidence to support the advantage of schema-based memory boosting. Moreover, the source of the effects needs to be further clarified using additional analyses.Concerns:1. Schema-based boost in memory and predictive eye movement lacks a control group who neither know the rule nor play the game before. The prediction is that the control group will show (1) worse memory; (2) their error patterns are different and unrelated to p (move).

Please see our response to Essential revisions comment #2.

2. Most of the correlation analyses mixed the variability levels and thus confused the source of the memory boosting effects. For example, when calculating the correlation results (Figure 2C), each data point represents a session of a participant. In this case, the source of the effect could come from either session-level variability that is due to training or participant-level variability. In fact, all the linear mixed-effect models have the same problem and need to be fixed. I would not suggest excluding the random effect for slopes since that would increase the probability of Type I error and as a consequence, the observed effect might arise from individual differences rather than experimental manipulation (see Brauer and Curtin, Psychol Methods, 2018 and Bates, arXiv, 2018 for guides). It is important to include individual-level random effects in the analysis.

Please see our response to Essential revisions comment #3.

Reviewer #2 (Recommendations for the authors):This paper by Huang and colleagues uses a simple board game, "4-in-a-row," to test the relationship between schemas and episodic memory. Specifically, the goal here was to examine the way schemas can guide encoding of sequential events as the game unfolded. In sum, the authors report that memory for move sequences in the game improved with player skill, and that memory was particularly improved for schema-consistent sequences. Further, eye movements during encoding were associated with stronger memory for move sequences, which was most prevalent in expert players.We think this study does a fine job of answering how the development of schema aid in prediction and thereby memory through use of a stimulus that is at once naturalistic but learnable by participants in laboratory time. This is a difficult set of boxes to check. While 4-in-a-row is certainly a simplistic way of testing encoding and retrieval of "events," it nonetheless tests memory in a fairly rich and meaningful way compared to much existing research, especially with regard to sequential order. The approach itself strikes a really nice balance between complexity in capturing elements of naturalistic human behavior, and simplicity in terms of analyses. Overall we think this experiment is sound, and the paper is well-written.We have some requests for expansion/clarification and a few suggestions for strengthening the paper.1. Looking at Figure 3c and 3d, it struck us that the slope of recall accuracy increases linearly across sessions 1-4, but then drops pretty markedly to sessions 5 and 6. If things simply leveled-off or dropped slightly, it would seem less important or interesting, but the drop-off is quite stark. What could be happening here? What do the authors make of this?

Please see our response to Essential revisions comment #4.

2. It was actually a bit surprising to us that the authors didn't use pupillometry when evaluating surprise. Pupil dilation has been strongly linked to the constructs of surprise and prediction error, more so than gaze. We are not necessarily suggesting that the authors rerun their model using pupillometry, but some rationale for why gaze rather than pupil dilation was optimal for testing surprise in this context would help. As it stands, we felt that this disconnect left the reasoning behind the surprise model feeling unclear, and we think a more explicit discussion of this would be helpful.

Here, we sought to operationalize surprise in terms of prediction accuracy. High surprise means that the specific move prediction made by the participant (as measured via gaze position) was not confirmed. We have now clarified our definition of surprise in the Results:

“The second measure is *surprise*, which is the negative log of the proportion of fixation time spent looking at the correct next move position. A high level of surprise indicates that a participant spent very little time looking at the square where the next move ended up appearing, which indicates low prediction accuracy.” (line 366-370)

For a discussion of pupil dilation effects in the study, please see our response to Essential revisions comment #4.

3. As the authors themselves flesh out in the Discussion, there is a fair amount of inconsistency in the literature as to whether schema-consistent (or "congruent," as others often say) or schema-inconsistent information provides relative a boost to episodic memory. And although we think their points related to this in the Discussion are fine, we think it would be worth laying out the reasoning for the prediction that schema-consistency should benefit memory in the Introduction. Our reasoning here is that, from our point of view, it could have easily gone either way. And we think that the field can benefit at this point from clear logic laying out predictions related to schema and memory.

Please see our response to Essential revisions comment #4.

4. Did the authors analyze reaction time data at all? To be clear, we are not making a strong suggestion to do so, but given links between surprise and response time variability, and the logical relationship between increasing experience and time taken to do something, it seems a logical thing to do in characterizing performance.

Please see our response to Essential revisions comment #4.

Reviewer #3 (Recommendations for the authors):This paper presents a series of analyses of the relationship between predictive schemas – representations that allow people to form predictions about events in their environment – and performance in a memory task. Participants in the experiments learned to play a novel board game over a period of several days, and also performed a memory task in which they had to recall sequences of board positions. The first time they performed the memory task they had no experience with the game, so these just looked like random configurations of black and white circles. As players became more familiar with the game, the memory performance also improved.Several analyses dig deeper into this phenomenon, examining how participants' memory for specific board positions are related to the probability of the corresponding moves in the game, as estimated by an AI agent. This analysis shows that people have better memory for higher probability moves. By bringing another group of participants into the laboratory and performing eye tracking, it was shown that people's eyes movements also seem to be influenced by the probability of moves. Further analyses that express confidence and surprise in terms of these eye movements – how focused people seem to be on a specific position and how long they spend looking at the correct position – show that these quantities are also related to memory.Overall, I think this is an impressive piece of work that does a nice job of exploring several questions about the nature and origins of schema-based effects on memory performance through the clever use of a novel task. However, I do think there is one substantial potential confound. As people are getting better at playing the game, it seems like one potential strategy they might use is to worry less about memorizing the board positions and instead to simply consider how they would play from that position. People following this strategy would seem to account for all of the results that are presented: it would result in a higher probability of being correct on higher probability moves, and potentially produce the associated effects involving eye-movements since those eye-movements tend to be concentrated on the high probability moves as well.This kind of confound is likely to arise in studies looking at schema effects on memory, because if people have a schema they can use that schema to make informed guesses that are more likely to be right for schema-consistent items. However, in this setting it seems like there is actually an opportunity to address this confound directly. I think this could be done by conducting a further study in which people play from the same board positions that are used to initiate the memory recall sequences – perhaps doing so on different days so as to remove concerns about interference. By measuring the actual play sequences that people generate, it would be possible to focus the analyses on cases where the memory sequences differ from these play sequences. In fact, if the play sequences are generated the day before the memory test, the memory test could include two kinds of trials: one set where the memory sequence corresponds to that produced by the participant on the previous day, and one where it diverges. If the reported effects are observed in both of these conditions, that helps to decouple simulated play from generic schema use (that is, if a move in a memory sequence that has high probability but wasn't played by the participant the previous day is still remembered well, it provides strong support for a schema effect rather than a play-based strategy).

Please see our response to Essential revisions comment #1.

[Editors' note: further revisions were suggested prior to acceptance, as described below.]

Reviewer #1 (Recommendations for the authors):About the two major comments or experiments I raised, the authors have done a fine job addressing them. They have performed a simulation analysis to exclude the alternative explanation raised in the first question. They chose to use the first session of the data as the control group to answer the second question.Their answers are fine but I would like to clarify my original point that may have been unclear. That is, showing the learning curve rather than a single point of the control group's performance, i.e., the first session, as a control.

Also, in response to our request for additional clarification, an editor wrote:

(1) providing the learning curve of a control group who are naïve to game rules, instead of using only the first session of the old data as a control. The authors chose to use the first session of data as a control group to answer the second question raised by Reviewer 1. Meanwhile, to demonstrate that the memory improvement after the first gameplay task is due to schema as claimed by the authors rather than training practice, it would be helpful to show the learning curve of a control group who are naïve to game rules throughout the whole experiment, rather than only the first-day memory performance.

Our response: Providing a full learning curve for a control group presents some logistical challenges, since this would require at least three months of new data collection using our department’s shared eye-tracking equipment. There is also a more fundamental challenge, in that participants are likely to eventually learn the rules of the game through pure exposure, even without explicit instruction. We now address these issues in the Methods:

“We did not include a control group that completed six memory sessions without being told the rules of the game or playing against the AI agent. Ensuring that participants never develop a game schema would be difficult, since even without explicit instruction they could still learn the schema through implicit statistical learning or by guessing the (simple) rules of the game. Due to these challenges, we instead used the first-session performance of each participant as our no-schema control.”

We also discuss the limitations of our one-session control in our Discussion:

“Note that some of the overall accuracy increase in the memory task across sessions could be attributable to a practice effect, but general practice-related improvement does not explain the schema-related effects that we observed on individual trials.”

Reviewer #4 (Recommendations for the authors):Huang et al. examined the influence of schemas on prediction and sequence memory for a novel board game. Participants' memory for sequences of moves was tested across several sessions as they learned the game and improved. Over the sessions, and accompanied by a skill increase, participants' memory for the sequences improved. This learning was also related to eye-movements during encoding which were predictive of memory for the sequences.This study does an excellent job of linking schema development to prediction/attention and memory.Comments:In the first paragraph of the Results section, it was unclear whether during the game play task participants were provided instructions or whether they had to learn the game through some kind of trial and error. It should be specified that they were provided with rules since early in the paragraph it notes that the participants were naive to the fact that a game was being shown in the initial memory test phase.

Thank you for your comments. We have now clarified this in the Results section.

“In a separate game play task (occurring after the memory task on all but the last day), the player was provided with the rules of the game and played the 4-in-a-row game against an AI opponent, staircased to match the skill level of the player.”

"Modeling schema-related memory improvement" In this section a model is trained based on moves from a near optimal AI. A probability distribution is made for upcoming moves based on the likelihood a move would be made by a near optimal AI. This probability is interpreted to be related to the schema consistency of a move. This may be true for an expert player. However, a near optimal move may not be schema-consistent for a new player or maybe even an average player. The participant's internal schema when they are beginners and progress may not map on to the near optimal AI moves. This is an interesting analysis, but calling the near optimal AI move schema consistent just slightly seems like a miswording for me.

Thank you for your comments on the use of the term "near-optimal AI" and its relation to schema consistency. Upon reflection, we realize that describing our gameplay model as "near-optimal" might be misleading. Our model was trained on near-optimal gameplay generated by near-optimal AI used in van Opheusden et al. (2021). However, our model is a simplified myopic model, which only captures some of the features of the near-optimal gameplay. We have removed the words “optimal” or “near-optimal” when they are used to describe our gameplay model.

We have clarified our use of “schema consistency” in the result section.

“Here, schema consistency is used as an objective, subject-independent measure of how good a move is. Each subject will exhibit different degrees of alignment to this “ground-truth” schema.”

We agree that individual participants’ schemas (especially those of the novices) will not always match exactly with our gameplay model. This difference in schema across people is one of the things that leads to the variability in our dependent measures (like how much schema-dependent prediction they exhibit). We would also like to note that although people have different degrees of alignment to the optimal AI schema, these schemas are qualitatively the same. We’ve included a discussion of this in Method section:

“We use the model’s evaluation of each move as a measure of the schema consistency of a move, assuming that stronger and weaker players have qualitatively the same schema for what makes a good move. An alternative possibility is that individuals develop schemas that are qualitatively distinct from the gameplay model, which will be reflected in some people playing moves that have very low probability in our gameplay model. We did not find evidence for this possibility in our dataset; during the gameplay sessions, all participants tended to play moves with relatively high probability under the gameplay model (*M* = 0.213, range = 0.143 – 0.297, compared to a random-move baseline of 0.040). We also observed a strong correlation between Elo and this move probability measure (*r* = .40, *p* <.001), showing that stronger players are more aligned with our model. Future work (with more extensive gameplay sessions) could attempt to model unique feature weights for each specific individual.”

Discussion "One possibility is that their study (in which participants implicitly learned statistical dependencies between image categories presented in a continuous stream) engages a predictive process that relies on the hippocampus (Schapiro et al., 2017), creating interference between prediction and encoding processes, while the explicit schematic predictions in our study could rely on a different network such as the medial prefrontal cortex " While this is a compelling assertion, can you describe anything that psychologically varies between these tasks? For example, Shapiro et al. had participants learn arbitrary statistical sequences, but here the sequences could be understood in the context of rules, goals, and progression towards goal states.

Thank you for your points about the cognitive difference between tasks. We have now incorporated these cognitive differences into our description of why they might rely on different neural mechanisms:

“One possibility is that their study (in which participants implicitly learned arbitrary statistical dependencies between image categories presented in a continuous stream) engages a predictive process that relies on the hippocampus (Schapiro et al., 2017), creating interference between prediction and encoding processes. On the other hand, the explicit schematic predictions in our study that are based on rules and progression towards goal states might rely on a different network such as the medial prefrontal cortex (Bonasia et al., 2018; Robin and Moscovitch, 2017; van Kesteren et al., 2012).”

Methods: During the memory task, 10 out of the 30 sequences ended in a win state. Since a win-state caps off a sequence this could be thought of as an event boundary. Was there any effect of these 10 sequences that differed from the other 20 or interacted with other effects? For example, if a win-state is an event boundary, is particularly salient, and is better remembered, then a participant merely has to reconstruct the sequence from probe to win-state, rather than remembering each move.

Thank you for your suggestions. We have now conducted additional analyses that indeed show that memory for sequences that ended in a win state differ from the rest of other sequences after people developed a schema. We discussed this finding in the Results:

“Out of the 30 sequences shown to participants in each session, ten of them ended with one player successfully getting four pieces in a row (more details in Methods). After learning the rules of the game, the presence of such a win could be a salient event for the participant and could lead to changes in memory performance. Indeed, we found that in sessions 2-6, memory for sequences that ended in a win state was significantly better (t = 6.67, p <.001). We did not observe this pattern in the first session (before participants were taught the game rules), and actually found a marginally significant effect in the opposite direction, with worse memory for winning sequences (t = -2.01, p = .052). This result provides additional evidence that schema-related features of a sequence play a role in memory performance.”

Modeling schema-related memory improvement" In this section the authors missed a period in "near- optimal AI agent The model identifies"

Fixed.